# THE REPRESENTATION JENSEN-SHANNON DIVERGENCE

## ABSTRACT

Statistical divergences quantify the difference between probability distributions, thereby allowing for multiple uses in machine-learning. However, a fundamental challenge of these quantities is their estimation from empirical samples since the underlying distributions of the data are usually unknown. In this work, we propose a divergence inspired by the Jensen-Shannon divergence which avoids the estimation of the probability density functions. Our approach embeds the data in a reproducing kernel Hilbert space (RKHS) where we associate data distributions with uncentered covariance operators in this representation space. Therefore, we name this measure the representation Jensen-Shannon divergence (RJSD). We provide an estimator from empirical covariance matrices by explicitly mapping the data to an RKHS using Fourier features. This estimator is flexible, scalable, differentiable, and suitable for minibatch-based optimization problems. Additionally, we provide an estimator based on kernel matrices without an explicit mapping to the RKHS. We provide consistency convergence results for the proposed estimator as well as connections with Shannon's differential entropy. Moreover, we demonstrate that this quantity is a lower bound on the Jensen-Shannon divergence, leading to a variational approach to estimate it with theoretical guarantees. We leverage the proposed divergence to train generative networks, where our method mitigates mode collapse and encourages samples diversity. Additionally, RJSD surpasses other state-of-the-art techniques in multiple two-sample testing problems, demonstrating superior performance and reliability in discriminating between distributions.

## 1 INTRODUCTION

Divergences quantify the difference between probability distributions. In machine-learning, divergences can be applied to a wide range of tasks, including generative modeling (generative adversarial networks, variational auto-encoders), two-sample testing, anomaly detection, and distribution shift detection. The family of $f$-divergences is among the most popular statistical divergences, including the well-known Kullback-Leibler and Jensen-Shannon divergences. A fundamental challenge to using divergences in practice is that the underlying distribution of data is unknown, and thus divergences must be estimated from observations. Several divergence estimators have been proposed (Yang & Barron, 1999; Sriperumbudur et al., 2012; Krishnamurthy et al., 2014; Moon & Hero, 2014; Singh & Póczos, 2014; Li & Turner, 2016; Noshad et al., 2017; Moon et al., 2018; Bu et al., 2018; Berrett & Samworth, 2019; Liang, 2019; Han et al., 2020; Sreekumar & Goldfeld, 2022), most of which fall into four categories: plug-in, kernel density estimation, $k$-nearest neighbors, and neural estimators.

Kernel methods are another approach for measuring the interaction between probability distributions. For example, the maximum mean discrepancy (MMD) (Gretton et al., 2012) is a divergence computed as the distance between the mean embeddings (first-order moments) of the two probability distributions in a reproducing kernel Hilbert space (RKHS). However, due to the underlying geometry, MMD lacks a straightforward connection with classical information theory tools (Bach, 2022). On the other hand, covariance operators (second-order moments) in RKHS have been used to propose multiple information theoretic quantities, such as marginal, joint, and conditional entropy (Sanchez Giraldo et al., 2014), as well as mutual information (Yu et al., 2019), and total correlation (Yu et al., 2021). However, strategies for estimating divergences within this framework have been less explored.

To fill this void, we propose a kernel-based information theoretic learning framework for divergence estimation. We make the following contributions:

- A novel divergence, the representation Jensen-Shannon divergence (RJSD), that avoids the estimation of the underlying density functions by mapping the data to an RKHS where distributions can be embedded using uncentered covariance operators acting in this representation space.

- An estimator from empirical covariance matrices that explicitly map data samples to an RKHS using Fourier features. This estimator is flexible, scalable, differentiable, and suitable for minibatch-based optimization problems. Additionally, an estimator based on kernel matrices without an explicit mapping to the RKHS is provided. Consistency results and sample complexity bounds for the proposed estimator are discussed.

- A connection between the kernel-based entropy and Shannon's entropy, as well as the relationship between RJSD with the classical Jensen-Shannon divergence. Namely, RJSD emerges as a lower bound on the classical Jensen-Shannon divergence enabling the construction of a variational estimator for the classical Jensen-Shannon divergence with statistical guarantees.

We use RJSD for training generative adversarial networks and show that it prevents mode collapse and encourages diversity, leading to more accurate and heterogeneous results. We also apply RJSD for two-sample testing problems and show that it accurately detects differences between probability distribution functions even for cases where other state-of-the-art measures fall short.

## 2 BACKGROUND

### 2.1 MEAN EMBEDDINGS AND COVARIANCE OPERATORS

Let $(\mathcal{X}, \mathbf{B}_{\mathcal{X}})$ be a measurable space and $\kappa : \mathcal{X} \times \mathcal{X} \to \mathbb{R}_{\geq 0}$ be a positive definite kernel. There exists a mapping $\phi : \mathcal{X} \to \mathcal{H}$, where $\mathcal{H}$ is a reproducing kernel Hilbert space (RKHS), such that $\kappa(x, x') = \langle \phi(x), \phi(x') \rangle_{\mathcal{H}}$. The kernel mean embedding (Smola et al., 2007) is a mapping $\mu$ from $\mathcal{M}_+^1(\mathcal{X})$ to $\mathcal{H}$, where $\mathcal{M}_+^1(\mathcal{X})$ is the space of probability measures on $\mathcal{X}$. The kernel mean embedding is defined as follows:

$$\mu_{\mathbb{P}} = \mathbb{E}_{X \sim \mathbb{P}}[\phi(X)] = \int_{\mathcal{X}} \phi(x) \, \mathrm{d}\, \mathbb{P}(x), \text{ for } \mathbb{P} \in \mathcal{M}_+^1. \tag{1}$$

An important property of the mean embedding is that if $\mathbb{E}_{X \sim \mathbb{P}}[\kappa(X, X)] < \infty$, for any $f \in \mathcal{H}$, then $\mathbb{E}_{X \sim \mathbb{P}}[f(X)] = \langle f, \mu_{\mathbb{P}} \rangle_{\mathcal{H}}$.

Another related mapping is the uncentered covariance operator (Baker, 1973). In this case, $\mathbb{P} \in \mathcal{M}_+^1$ is mapped to an operator $C_{\mathbb{P}} : \mathcal{H} \to \mathcal{H}$ given by:

$$C_{\mathbb{P}} = \mathbb{E}_{X \sim \mathbb{P}}[\phi(X) \otimes \phi(X)] = \int_{\mathcal{X}} \phi(x) \otimes \phi(x) \, \mathrm{d}\, \mathbb{P}(x), \tag{2}$$

where $\otimes$ is the tensor product. Similarly, for any $f, g \in \mathcal{H}$, $\mathbb{E}_{X \sim \mathbb{P}}[f(X)g(X)] = \langle g, C_{\mathbb{P}}f \rangle_{\mathcal{H}}$. The covariance operator is positive semi-definite and Hermitian (self-adjoint). Additionally, if the kernel is bounded, the covariance operator is trace class (Sanchez Giraldo et al., 2014; Bach, 2022). The spectrum of the covariance operator is discrete and consists of non-negative eigenvalues $\lambda_i$ with $\sum \lambda_i < \infty$ for which we can extend functions on $\mathbb{R}$ such as $t \log(t)$ and $t^{\alpha}$ to covariance operators via their spectrum Naoum & Gittan (2004). For a sample $\mathbf{X} = \{x_i\}_{i=1}^N$ of size $N$, where $x_i \in \mathcal{X}$, drawn from $\mathbb{P}$, the empirical uncentered covariance operator is defined as:

$$\boldsymbol{C}_{\mathbf{X}} = \frac{1}{N} \sum_{i=1}^{N} \phi(x_i) \otimes \phi(x_i) \tag{3}$$

### 2.2 KERNEL-BASED INFORMATION THEORY

We can define information theoretic quantities on the spectrum of normalized covariance operators with unit trace. This observation was made by Sanchez Giraldo et al. (2014) who proposed the kernel-based entropy functional: $S_{\alpha}(C_{\mathbb{P}}) = \frac{1}{1-\alpha} \log \left[ \mathrm{Tr}(C_{\mathbb{P}}^{\alpha}) \right]$. $\mathrm{Tr}(\cdot)$ denotes the trace operator, $C_{\mathbb{P}}^{\alpha}$ is defined based on the spectrum of $C_{\mathbb{P}}$, and $\alpha > 0$ is the entropy order. This quantity resembles quantum Rényi entropy (Müller-Lennert et al., 2013) where the covariance operator plays the role of a density matrix [1]. In the limit when $\alpha \to 1$, $S_{\alpha \to 1}(C_{\mathbb{P}}) = - \mathrm{Tr}(C_{\mathbb{P}} \log C_{\mathbb{P}})$ becomes von Neumann entropy of the covariance operator. This connection between covariance operators in RKHS and information theory has been also discussed by Bach (2022).

---

[1] A density matrix is a matrix that describes the quantum state of a physical system

**Kernel-based entropy estimator:** The kernel-based entropy estimator relies on the spectrum of the empirical uncentered covariance operator in Eqn. 3. We focus on the case of normalized kernels where $\kappa(x, x) = 1$ for all $x \in \mathcal{X}$. We denote the Gram matrix $\mathbf{K_X}$, consisting of all normalized pairwise kernel evaluations of data points in the sample $\mathbf{X}$, that is $(\mathbf{K_X})_{ij} = \kappa(x_i, x_j)$ for $i, j = 1, \dots, N$. It can be shown that $C_\mathbf{X}$ and $\frac{1}{N}\mathbf{K_X}$ have the same non-zero eigenvalues (Sanchez Giraldo et al., 2014; Bach, 2022) yielding the kernel-based entropy estimator:

$$S\left(\mathbf{K_X}\right) = -\operatorname{Tr}\left(\tfrac{1}{N}\mathbf{K_X}\log\tfrac{1}{N}\mathbf{K_X}\right) = -\sum_{i=1}^{N}\lambda_i\log\lambda_i, \tag{4}$$

where $\lambda_i$ represents the $i$th eigenvalue of $\frac{1}{N}\mathbf{K_X}$. The eigen-decomposition of $\mathbf{K_X}$ has $\mathcal{O}(N^3)$ time complexity, which needs to be taken into consideration depending on the use case.

**Covariance-based estimator:** Alternatively, we can use an explicit mapping $\phi_\omega : \mathcal{X} \to \mathcal{H}^D$ to a finite dimensional RKHS. We propose to use Fourier features to construct a mapping function to $\mathcal{H}^D$. For $\mathcal{X} \subseteq \mathbb{R}^d$ and a shift-invariant kernel $\kappa(x, x') = \kappa(x - x')$, the random Fourier features (RFF) (Rahimi & Recht, 2007) is a method to create a smooth feature mapping $\phi_\omega(x) : \mathcal{X} \to \mathbb{R}^D$ so that $\kappa(x - x') \approx \langle\phi_\omega(x), \phi_\omega(x')\rangle$. To generate an RFF mapping, we compute the Fourier transform of the kernel, $p(\boldsymbol{\omega}) = \frac{1}{2\pi}\int e^{-j\boldsymbol{\omega}^\top\boldsymbol{\delta}}\kappa(\boldsymbol{\delta})d\boldsymbol{\delta}$, which yields a distribution on $\mathbb{R}^d$ with density $p(\boldsymbol{\omega})$. From this distribution, we draw $\frac{D}{2}$ i.i.d samples $\boldsymbol{\omega}_1, \dots, \boldsymbol{\omega}_{D/2} \in \mathbb{R}^d$. Finally, the mapping is given by $\phi_\omega(\boldsymbol{x}) = \sqrt{\frac{2}{D}}\left[\cos(\boldsymbol{\omega}_1^\top\boldsymbol{x}), \sin(\boldsymbol{\omega}_1^\top\boldsymbol{x}), \cdots, \cos(\boldsymbol{\omega}_{D/2}^\top\boldsymbol{x}), \sin(\boldsymbol{\omega}_{D/2}^\top\boldsymbol{x})\right]$.

Letting $\boldsymbol{\Phi_X} = \left[\phi_\omega(\boldsymbol{x_1})^T, \phi_\omega(\boldsymbol{x_2}), \cdots, \phi_\omega(\boldsymbol{x_N})\right]^T$ be the $N \times D$ matrix containing the mapped samples, we can compute the empirical uncentered covariance matrix as $\boldsymbol{C_X} = \frac{1}{N}\boldsymbol{\Phi}_X^\top\boldsymbol{\Phi}_X$. Finally, we exploit the eigenvalues of the uncentered covariance matrix to compute the von Neumann entropy of $\boldsymbol{C_X}$ as:

$$S\left(\boldsymbol{C_X}\right) = -\operatorname{Tr}\left(\boldsymbol{C_X}\log\boldsymbol{C_X}\right) = -\sum_{i=1}^{D}\lambda_i\log\lambda_i, \tag{5}$$

where $\lambda_i$ represents the $i$th eigenvalue of $\boldsymbol{C_X}$. This eigendecomposition has $\mathcal{O}(D^3)$ time complexity, where $D$ is independent of the sample size.

Both estimators of kernel-based entropy can be used in gradient based learning (Sanchez Giraldo & Principe, 2013; Sriperumbudur & Szabó, 2015). The kernel-based entropy has been used as a building block for other matrix-based measures, such as joint and conditional entropy, mutual information (Yu et al., 2019), total correlation (Yu et al., 2021), and divergence (Hoyos Osorio et al., 2022). Despite the success of the aforementioned measures, their connection with the classical information theory counterparts remains unclear.

For the case where $\mathcal{X} \subseteq \mathbb{R}^d$ and the distribution $\mathbb{P}$ has a corresponding probability density function $p$, we can establish an explicit connection between the kernel-based entropy estimator and Shannon's differential entropy, $H(p) = -\int_\mathcal{X} p(x)\log p(x)dx$.

**Definition 1.** *Let $\phi : \mathcal{X} \to \mathcal{H}$ be a mapping to a reproducing kernel Hilbert space (RKHS), and $\kappa : \mathcal{X} \times \mathcal{X} \to \mathbb{R}_{\geq 0}$ be a positive definite kernel, such that $\kappa(x, x') = \langle\phi(x), \phi(x')\rangle_\mathcal{H}$, and $\kappa(x, x) = 1$ for all $x \in \mathcal{X}$. Then, the **kernel density function** induced by the mapping $\phi$ is defined as follows:*

$$\hat{p}(x) = \frac{1}{h}\langle\phi(x), C_\mathbb{P}\phi(x)\rangle = \frac{1}{h}\int_\mathcal{X}\kappa^2(x, x')d\mathbb{P}(x') = \frac{1}{h}\int_\mathcal{X}\kappa^2(x, x')p(x')dx', \tag{6}$$

*where $h = \int_\mathcal{X}\langle\phi(x), C_\mathbb{P}\phi(x)\rangle\,dx$ is the normalizing constant.*

Eqn. 6 can be interpreted as an instance of the Born rule which calculates the probability of finding a state $\phi(x)$ in a system described by the covariance operator $C_\mathbb{P}$ (González et al., 2022). Equivalently, the right-most side can be seen as smoothing the density $p$ with a kernel $\kappa^2(\cdot, \cdot)$.

**Theorem 1.** *Let $\hat{p}(x)$ be the kernel density function induced by a mapping $\phi : \mathcal{X} \to \mathcal{H}$, then, the cross entropy between $p$ and $\hat{p}$ is:*

$$H(p, \hat{p}) = -\int_\mathcal{X} p(x)\log\hat{p}(x)dx = S(C_\mathbb{P}) + \log(h). \tag{7}$$

*Proof:* See Appendix A.1.

From Theorem 1 we can see that the covariance operator entropy relates to a plug-in estimator of Shannon's differential entropy based on the Parzen density estimator. We can use well-known results about the convergence of the Parzen-density estimator (Dmitriev & Tarasenko, 1974) to derive the convergence of both kernel-based and covariance-based entropy estimators.

**Theorem 2.** *let $\kappa(x, x') = \exp\left(-\gamma_N \|x - x'\|^2\right)$ be a Gaussian kernel with scale parameter $\gamma_N = \frac{1}{2} N^{1/4}$, and let $p(x)$ be any bounded probability density function on $\mathcal{X}$, then $S(\mathbf{K_X})$ converges to $H(p)$ as $N \to \infty$ with probability one. Proof:* See Appendix A.2.

A similar result can be proved for empirical covariance operators generated through RFFs.

**Theorem 3.** *Let $\phi_\omega : \mathcal{X} \to \mathbb{R}^D$ be a Fourier features mapping approximating the Gaussian kernel with scale parameter $\frac{\gamma_N}{2} = \frac{1}{4} N^{1/4}$, and let $p(x)$ be any bounded probability density function on $\mathcal{X}$. Then, $S(\boldsymbol{C}_\mathbf{X})$ converges to $H(p)$ as $N \to \infty$ and $D \to \infty$ with probability one. Proof:* See Appendix A.3.

## 3 REPRESENTATION JENSEN-SHANNON DIVERGENCE

For two probability measures $\mathbb{P}$ and $\mathbb{Q}$ on a measurable space $(\mathcal{X}, \mathbf{B}_\mathcal{X})$, the Jensen-Shannon divergence (JSD) is defined as follows:

$$D_{JS}(\mathbb{P}, \mathbb{Q}) = H\left(\frac{\mathbb{P} + \mathbb{Q}}{2}\right) - \frac{1}{2}\left(H(\mathbb{P}) + H(\mathbb{Q})\right), \tag{8}$$

where $\frac{\mathbb{P}+\mathbb{Q}}{2}$ is the mixture of both distributions and $H(\cdot)$ is Shannon's entropy. Properties of JSD, such as boundedness, convexity, and symmetry have been extensively studied (Briët & Harremoës, 2009; Sra, 2021). The Quantum counterpart of the Jensen-Shannon divergence (QJSD) between density matrices $\rho$ and $\sigma$ is defined as $D_{JS}(\rho, \sigma) = S\left(\frac{\rho+\sigma}{2}\right) - \frac{1}{2}\left(S(\rho) + S(\sigma)\right)$, where $S(\cdot)$ is von Neumann's entropy. QJSD is everywhere defined, bounded, symmetric, and positive if $\rho \neq \sigma$ (Sra, 2021). Similar to the kernel-based entropy, we let the covariance operators play the role of the density matrices to derive a measure of divergence that can be computed directly from data samples.

**Definition 2.** *Let $\mathbb{P}$ and $\mathbb{Q}$ be two probability measures defined on a measurable space $(\mathcal{X}, \mathbf{B}_\mathcal{X})$, and let $\phi : \mathcal{X} \to \mathcal{H}$ be a mapping to a reproducing kernel Hilbert space (RKHS) $\mathcal{H}$, such that $\langle\phi(x), \phi(x)\rangle_\mathcal{H} = 1$ for all $x \in \mathcal{X}$. Then, the **representation Jensen-Shannon divergence** (RJSD) between uncentered covariance operators $C_\mathbb{P}$ and $C_\mathbb{Q}$ is defined as:*

$$D_{JS}^\phi(C_\mathbb{P}, C_\mathbb{Q}) = S\left(\frac{C_\mathbb{P} + C_\mathbb{Q}}{2}\right) - \frac{1}{2}\left(S(C_\mathbb{P}) + S(C_\mathbb{Q})\right). \tag{9}$$

### 3.1 THEORETICAL PROPERTIES

RJSD inherits most of the properties of classical and quantum Jensen-Shannon divergence. *Non-negativity*: $D_{JS}^\phi(C_\mathbb{P}, C_\mathbb{Q}) \geq 0$. *Positivity*: $D_{JS}^\phi(C_\mathbb{P}, C_\mathbb{Q}) = 0$ if and only if $C_\mathbb{P} = C_\mathbb{Q}$. *Symmetry*: $D_{JS}^\phi(C_\mathbb{P}, C_\mathbb{Q}) = D_{JS}^\phi(C_\mathbb{P}, C_\mathbb{Q})$. *Boundedness*: $D_{JS}^\phi(C_\mathbb{P}, C_\mathbb{Q}) \leq \log(2)$. Also, $D_{JS}^\phi(C_\mathbb{P}, C_\mathbb{Q})^{\frac{1}{2}}$ is a metric on the cone of uncentered covariance matrices in any dimension (Virosztek, 2021).

Below, we introduce key properties of RJSD and the connection with its classical counterpart.

**Theorem 4.** *For all probability measures $\mathbb{P}$ and $\mathbb{Q}$ defined on $\mathcal{X}$, and covariance operators $C_\mathbb{P}$ and $C_\mathbb{Q}$ with RKHS mapping $\phi(\cdot)$ under the conditions of Definition 2, the following inequality holds:*

$$D_{JS}^\phi(C_\mathbb{P}, C_\mathbb{Q}) \leq D_{JS}(\mathbb{P}, \mathbb{Q}) \tag{10}$$

*Proof:* See Appendix A.4.

**Theorem 5.** *let $\mathbb{P}$ and $\mathbb{Q}$ be two probability measures defined on $\mathcal{X}$, with probability density functions $p$ and $q$ respectively. If there exists a mapping $\phi^*$ such that $p(x) = \frac{1}{h_\mathbb{P}} \langle\phi^*(x), C_\mathbb{P}\phi^*(x)\rangle$ and $q(x) = \frac{1}{h_\mathbb{Q}} \langle\phi^*(x), C_\mathbb{Q}\phi^*(x)\rangle$, then:*

$$D_{JS}(\mathbb{P}, \mathbb{Q}) = D_{JS}^{\phi^*}(C_\mathbb{P}, C_\mathbb{Q}). \tag{11}$$

*Proof:* See Appendix A.5.

This theorem implies that the bound in Eqn. 10 is tight for optimal functions $\phi(x)$ that approximate the true underlying distributions through Eqn. 6. Theorems 4 and 5 can be used to obtain a variational estimator of Jensen-Shannon divergence (see Section 4).

Finally, we show that RJSD relates to MMD with kernel $\kappa^2(\cdot, \cdot)$, where MMD is formally defined as $\mathrm{MMD}_\kappa(\mathbb{P}, \mathbb{Q}) = \|\mu_\mathbb{P} - \mu_\mathbb{Q}\|_{\mathcal{H}}$.

**Theorem 6.** *For all probability measures $\mathbb{P}$ and $\mathbb{Q}$ defined on $\mathcal{X}$, and covariance operators $C_\mathbb{P}$ and $C_\mathbb{Q}$ with RKHS mapping $\phi(x)$ such that $\langle \phi(x), \phi(x) \rangle_{\mathcal{H}} = 1 \quad \forall x \in \mathcal{X}$:*

$$D_{JS}^\phi(C_\mathbb{P}, C_\mathbb{Q}) \geq \frac{1}{8} \mathrm{MMD}_{\kappa^2}(\mathbb{P}, \mathbb{Q}) \tag{12}$$

*Proof:* See Appendix A.6.

The result of Theorem 6 should not be underestimated. Since MMD is a lower bound on the RJSD, any discrepancies between distributions that can be detected with MMD should be also detected with RJSD. That is RJSD should be at least as good as MMD. Moreover, it also shows that RJSD is well defined for characteristic kernel, for which RJSD is non zero if $\mathbb{P} \neq \mathbb{Q}$.

### 3.2 ESTIMATING THE REPRESENTATION JENSEN-SHANNON DIVERGENCE

Given two sets of samples $\mathbf{X} = \{x_i\}_{i=1}^N \subset \mathcal{X}$ and $\mathbf{Y} = \{y_i\}_{i=1}^M \subset \mathcal{X}$ with unknown distributions $\mathbb{P}$ and $\mathbb{Q}$, we propose two estimators of RJSD.

**Kernel-based estimator:** Here, we propose an estimator of RJSD from kernel matrices without an explicit mapping to the RKHS.

**Lemma 1.** *Let $\mathbf{Z}$ be the mixture of the samples of $\mathbf{X}$ and $\mathbf{Y}$, that is, $\mathbf{Z} = \{\mathbf{z}_i\}_{i=1}^{N+M}$ where $\mathbf{z}_i = \mathbf{x}_i$ for $i \in \{1, \ldots, N\}$ and $\mathbf{z}_i = \mathbf{y}_{i-N}$ for $i \in \{N+1, \ldots, N+M\}$. Also, let $\mathbf{K}_\mathbf{Z}$ be the kernel matrix consisting of all normalized pairwise kernel evaluations of the samples in $\mathbf{Z}$, then $S\left(\frac{N}{N+M} \boldsymbol{C}_\mathbf{X} + \frac{M}{N+M} \boldsymbol{C}_\mathbf{Y}\right) = S(\mathbf{K}_\mathbf{Z})$. (Proof: See Appendix A.7).*

Since the spectrum of $\mathbf{K}_\mathbf{X}$ and $\boldsymbol{C}_\mathbf{X}$ have the same non-zero eigenvalues, likewise $\mathbf{K}_\mathbf{Y}$ and $\boldsymbol{C}_\mathbf{Y}$, the divergence can be directly computed from samples in the input space as:

$$D_{JS}^\kappa(\mathbf{X}, \mathbf{Y}) = S(\mathbf{K}_\mathbf{Z}) - \left(\frac{N}{N+M} S(\mathbf{K}_\mathbf{X}) + \frac{M}{N+M} S(\mathbf{K}_\mathbf{Y})\right) \tag{13}$$

Leveraging the convergence results in Bach (2022)[Proposition 7] of the empirical estimator $S(\mathbf{K}_\mathbf{X})$ to $S(C_\mathbb{P})$, we can show that $D_{JS}^\kappa(\mathbf{X}, \mathbf{Y})$ converges to the population quantity $D_{JS}^\phi(C_\mathbb{P}, C_\mathbb{Q})$ at a rate $\mathcal{O}\left(\frac{1}{\sqrt{N}}\right)$, assuming $N = M$. Details of this rate are given in Appendix A.8. Additionally, a direct consequence of Theorem 2 is that under the same assumptions of the theorem, $D_{JS}^\kappa(\mathbf{X}, \mathbf{Y})$ converges to $D_{JS}(\mathbb{P}, \mathbb{Q})$ as $N \to \infty$ with probability one.

**Covariance-based estimator:** We propose to use Fourier features to construct a mapping function $\phi_\omega : \mathcal{X} \to \mathcal{H}_D$ to a finite-dimensional RKHS as explained in Section 2.2. Let $\boldsymbol{\Phi}_X \in \mathbb{R}^{N \times D}$ and $\boldsymbol{\Phi}_Y \in \mathbb{R}^{M \times D}$ be the matrices containing the mapped samples of each distribution. Then, the empirical uncentered covariance matrices are computed as $\boldsymbol{C}_\mathbf{X} = \frac{1}{N} \boldsymbol{\Phi}_X^\top \boldsymbol{\Phi}_X$ and $\boldsymbol{C}_\mathbf{Y} = \frac{1}{M} \boldsymbol{\Phi}_Y^\top \boldsymbol{\Phi}_Y$. Finally, the covariance-based RJSD estimator is defined as:

$$D_{JS}^\omega(\boldsymbol{C}_\mathbf{X}, \boldsymbol{C}_\mathbf{Y}) = S\left(\frac{N}{N+M} \boldsymbol{C}_\mathbf{X} + \frac{M}{N+M} \boldsymbol{C}_\mathbf{Y}\right) - \left(\frac{N}{N+M} S(\boldsymbol{C}_\mathbf{X}) + \frac{M}{N+M} S(\boldsymbol{C}_\mathbf{Y})\right), \tag{14}$$

Finally, we use Eqn. 5 to estimate the entropies of the covariance matrices. Notice, that the use of the Fourier features is not solely to reduce computational burden by approximating the kernel-based estimator. The Fourier features allow a parameterization of the representation space, for kernel-learning. We can treat the Fourier features as learnable parameters within a neural network (Fourier Feature network), optimizing them to maximize divergence and enhance its discriminatory power. Consequently, the Fourier features approach offers a more versatile estimator that extends beyond reducing computational cost.

## 4 VARIATIONAL ESTIMATION OF CLASSICAL JENSEN-SHANNON DIVERGENCE

We exploit the lower bound in Theorem 4 to derive a variational method for estimating the classical Jensen-Shannon divergence (JSD) given only samples from $\mathbb{P}$ and $\mathbb{Q}$. Accordingly, we choose $\Phi$ to be the family of functions $\phi_\omega : \mathcal{X}^d \to \mathcal{H}^D$ parameterized by $\omega \in \Omega$. Here, we aim to optimize the Fourier features to maximize the lower bound in Eqn. 4. Notice that we can also use a neural network $f_\omega$ with a Fourier features mapping $\phi_\omega$ in the last layer, that is, $\phi_\omega \circ f_\omega = \phi_\omega(f_\omega(x))$. We call this network a *Fourier-features network (FFN)*. Finally, we can compute the divergence based on this representation, leading to a neural estimator of classical JSD.

**Definition 3.** *(Jensen-Shannon divergence variational estimator). Let* $\Phi = \{\phi_\omega \circ f_\omega\}_{\omega \in \Omega}$ *be the set of functions parameterized by a FFN. We define our JSD variational estimator as:*

$$\widehat{D_{JS}}(\mathbb{P}, \mathbb{Q}) = \sup_{\omega \in \Omega} D_{JS}^\omega(C_\mathbb{P}, C_\mathbb{Q}). \tag{15}$$

This approach leverages the expressive power of deep networks and combines it with the capacity of kernels to embed distributions in a RKHS. This formulation allows to model distributions with complex structures and improve the convergence of the estimator by the universal approximation properties of the neural networks (Wilson et al., 2016; Liu et al., 2020). Algorithm 1 in Appendix B describes the proposed estimator.

## 5 EXPERIMENTS

### 5.1 VARIATIONAL JENSEN-SHANNON DIVERGENCE ESTIMATION

First, we evaluate the performance of our variational estimator of Jensen-Shannon divergence (JSD) in a tractable toy experiment. Here, $\mathbb{P} \sim p(x; l_p, s_p)$ and $\mathbb{Q} \sim p(x; l_q, s_q)$ are two Cauchy distributions with location parameters $l_p$ and $l_q$ and scale parameters $s_p = s_q = 1$. We vary the location parameter of $\mathbb{Q}$ over time to control the target divergence. We use a closed form of the JSD between Cauchy distributions derived by Nielsen & Okamura (2022) to determine the location parameter (see Appendix C.1 for more details). Then, we apply Algorithm 1 to estimate JSD drawing $N = 512$ samples from both distributions at every epoch. We compare the estimates of divergence against different neural estimators. JSD corresponds to the mutual information between the mixture distribution and a Bernoulli distribution indicating when a sample is drawn from $\mathbb{P}$ or $\mathbb{Q}$. Therefore, we use mutual information estimators to approach the JSD estimation, such as NWJ (Nguyen et al., 2010), infoNCE (Oord et al., 2018), CLUB (Cheng et al., 2020), MINE (Belghazi et al., 2018). We also employ KNIFE (Pichler et al., 2022) to estimate the entropy terms and compute JSD.

Fig. 1 shows the estimation results. All compared methods approximate JSD; however, some of them struggle to adapt to distribution changes. These abrupt adjustments could lead to instabilities during training. In contrast to the compared methods, the RJSD estimator accurately estimates divergence with a lower variance, adjusting itself smoothly to changes in the distributions. Additionally, by using Exponential Moving averages (EMA) of the covariance matrices, the estimation variance decreases further yielding a smoother estimation. Finally, we compute RJSD for a fixed set of Fourier features without any optimization (no gradients backpropagated), and we can observe that RJSD still approximates the true divergence. This result agrees with theorem 5 suggesting that the computed kernel implicitly approximates the underlying distributions of the data.

### 5.2 GENERATIVE ADVERSARIAL NETWORKS

Generative Adversarial Networks (GANs) are a family of models to generate images/audio. GANs algorithms minimize the dissimilarity between the generated and the real data distributions (Farnia & Ozdaglar, 2020). For example, the vanilla GAN algorithm (Goodfellow et al., 2020) minimizes the Jensen-Shannon divergence (JSD), whereas Wasserstein-GANs (Arjovsky et al., 2017) and MMD-GANs (Li et al., 2017) minimize their respective statistical distances.

GANs, however, usually suffer from mode collapse failing to cover the multiple modes (classes) of the real data (Choi & Han, 2022). This deficiency yields generative distributions with lower entropy compared to the target distribution (Che et al., 2016). One common approach to prevent mode collapse is through entropy regularizers (Belghazi et al., 2018; Dieng et al., 2019).

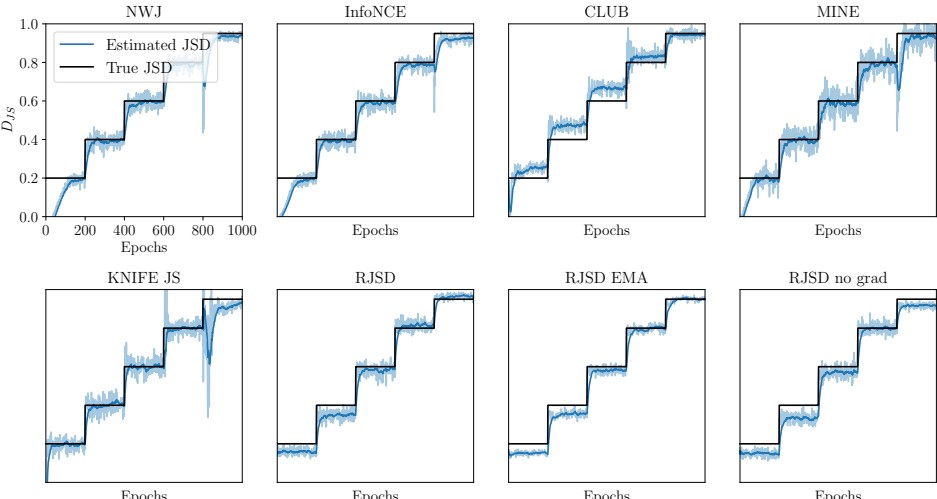

Figure 1: Jensen-Shannon Divergence estimation for two set of samples following Cauchy distributions (N = 512). We compare the following estimators: NWJ (Nguyen et al., 2010), infoNCE (Oord et al., 2018), CLUB (Cheng et al., 2020), MINE (Belghazi et al., 2018), KNIFE (Pichler et al., 2022), RJSD, RJSD with EMA, RJSD for a fixed kernel. The black line is the closed-form JS divergence between the Cauchy distributions. The parameters of the distributions are changed every 200 epochs to increase the divergence.

Below, we propose a methodology for training GANs using RJSD in the objective function. From first principles, RJSD should work for reducing mode collapse without requiring auxiliary entropy regularizers. The RJSD-GAN is formulated as follows:

$$\min_{\theta \in \Theta} \max_{\omega \in \Omega} D_{JS}^{\omega}(\mathbf{X}, \mathbf{Y}^{\theta}), \tag{16}$$

where $\mathbf{X}$ are samples from the real data, and $\mathbf{Y}^{\theta}$ are samples created by a generator $G_{\theta}$. Instead of classifying real and fake samples, we use a *Fourier-features network* $\{\phi_{\omega} \circ f_{\omega}\}_{\omega \in \Omega}$ (FFN, see Section 4) to learn a multidimensional representation in an RKHS where the divergence is maximized. Subsequently, the generator $\{G_{\theta}\}_{\theta \in \Theta}$ attempts to minimize RJSD. We follow a single-step alternating gradient method (see Algorithm 3 in Appendix B). We assess our GAN formulation in two well-known mode-collapse experiments: eight Gaussians dataset and stacked MNIST.

### 5.2.1 SYNTHETIC EXPERIMENTS

We apply RJSD to train a GAN in a synthetic experiment. The target distribution is a mixture of eight Gaussian distributions arranged in a circle. Fig. 2 shows the real data and the samples generated by various learning functions to train GANs. As expected, the standard (vanilla) GAN fails to generate samples from all modes (Fig. 2(a)). The Hinge (Lim & Ye, 2017) and Wasserstein-GP GANs (Gulrajani et al., 2017) successfully produce samples representing all eight modes, yet Figs. 2(b) and 2(c) exhibit generated samples with reduced variance/diversity (lower entropy) within each mode: a phenomenon termed intra-class collapse. As we observe, the generated samples fail to cover the entire support of each Gaussian mode clustering towards the center. In contrast to the compared methods, the samples generated by the RJSD-GAN show improved mode coverage and higher diversity. This is visually noticeable in Fig.

Table 1: KL divergence between real and generated distributions on eightmodes dataset.

| Average KL divergence | | |
| --- | --- | --- |
| RJSD | Wasserstein-GP | Hinge |
| **0.699 ± 0.245** | 0.981 ± 0.701 | 1.623 ± 1.000 |

Table 2: Number of modes and KL divergence between real and generated distributions on stacked MNIST.

| | Modes (Max 1000) | KL |
| --- | --- | --- |
| DCGAN (Radford et al., 2015) | 99.0 | 3.40 |
| ALI (Dumoulin et al., 2016) | 16.0 | 5.40 |
| Unrolled GAN (Metz et al., 2016) | 48.7 | 4.32 |
| VEEGAN (Srivastava et al., 2017) | 150 | 2.95 |
| WGAN-GP (Gulrajani et al., 2017) | 959.0 | 0.72 |
| PresGAN (Dieng et al., 2019) | 999.6 ± 0.4 | 0.11 ± 7.0e−2 |
| PacGAN (Lin et al., 2018) | 1000.0 ± 0 | 0.06 ± 1.0e−2 |
| GAN+MINE (Belghazi et al., 2018) | 1000.0 ± 0 | 0.05 ± 6.9e−3 |
| **GAN + rep JSD** | 1000.0 ± 0 | **0.04 ± 1.2e−3** |

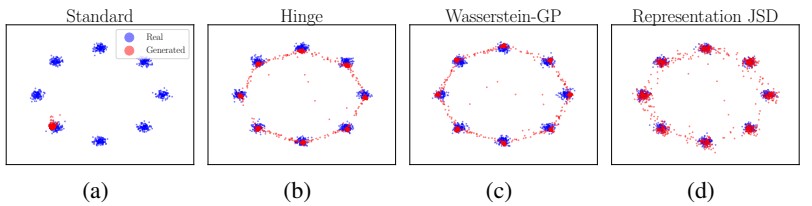

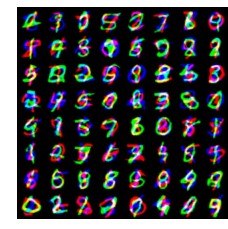

Figure 2: GANs with different loss functions to evaluate mode collapse in eight Gaussians dataset. RJSD improves mode coverage and sample diversity.

Figure 3: Generated samples using rep JSD.

2(d). Additionally, we perform the following quantitative analysis. We cluster the eight modes generated by each method and estimate their mean and covariance matrices (see Fig. 1 in Appendix C.2.1). Then, we calculate the Kullback-Leibler (KL) divergence between the real Gaussian modes and their generated counterparts. Finally, we average the divergence among the eight modes. Table 1 highlights the superiority of RJSD in terms of KL divergence when contrasted with the baseline methods. This empirical evidence supports the efficacy of RJSD to avoid mode collapse and to generate samples matching the target distribution beyond visual comparability.

### 5.2.2 STACKED MNIST

We conduct a quantitative evaluation to assess the efficacy of RJSD in reducing mode collapse on the stacked MNIST dataset. This dataset consists of three randomly sampled MNIST digits stacked along different color channels. This procedure results in 1000 possible classes (modes) corresponding to all combinations of the 10 digits. We use the standard DCGAN generator architecture (Radford et al., 2015), and modify the discriminator architecture to include a Fourier-features mapping (see implementation details in Appendix C.2.2). We compare our method against a considerable number of GAN algorithms using the same generator and following the same evaluation protocol. We utilize a pre-trained classifier to quantify the number of distinct generated modes. Additionally, we calculate the Kullback-Leibler (KL) divergence between the distribution of the generated modes and the real mode distribution. Finally, we average the results over five runs. Table 2 shows the results, and RJSD captures all modes and steadily generates samples from all classes achieving the lowest KL-divergence compared to the baseline approaches. Although our algorithm is a standard GAN that explicitly minimizes the Jensen-Shannon divergence, RJSD does not require the incorporation of entropy regularizers or mode-collapse prevention mechanisms beyond the learning function itself.

### 5.3 TWO SAMPLE TESTING

We evaluate the performance of RJSD for two-sample testing on different datasets and compare it against different state-of-the-art (SOTA) methods. We perform the following tests: (a) RJSD-FF: Two-sample test based on RJSD, optimizing the Fourier features. (b) RJSD-RFF: Two-sample test based on RJSD using random Fourier features, optimizing just the length-scale of the associated Gaussian kernel. (c) RJSD-D: Two-sample test based on RJSD using a deep Fourier-features network as explained in section 4 (see implementation details in Appendix C.3). (d) RJSD-K[2]: Two-sample test based on the kernel RJSD estimator, optimizing the length-scale of a Gaussian kernel. (e) MMD-O: Two-sample test based on MMD, optimizing the length-scale of the Gaussian kernel (Liu et al., 2020). (f) MMD-D: Two-sample test based on MMD with a deep kernel (Liu et al., 2020). (g) C2ST-L: a classifier two-sample test based on the output classification scores (Cheng & Cloninger, 2022). (h) C2ST-S: a classifier two-sample test based on the sign of the output classification scores (Lopez-Paz & Oquab, 2016).

We perform two-sample testing on two synthetic and two real-world datasets. Specifically, we perform permutation tests and the testing procedure is detailed in Appendix C.3.

**Blobs dataset (Liu et al., 2020):** In this dataset, $\mathbb{P}$ and $\mathbb{Q}$ are mixtures of nine Gaussians with the same modes. Each mode in $\mathbb{P}$ is an isotropic Gaussian; however, the modes in $\mathbb{Q}$ have different

---

[2]We did not perform this test for large size datasets due to computational restrictions

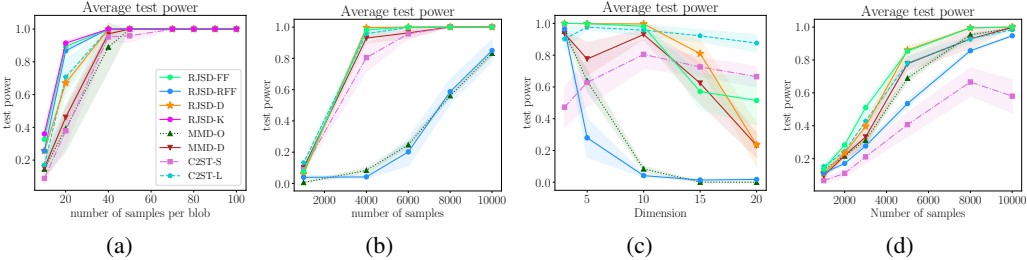

Figure 4: Average test power. (a) Blobs data. (b) High dimensional Gaussian mixture (GM), fixed $d = 10$. (c) High dimensional GM, fixed $N + M = 4000$ (d) Higgs data. Significance level $\alpha = 0.05$.

Table 3: MNIST average test power ($\alpha = 0.05$). Bold represents higher mean per column.

| $N + M$ | 200 | 300 | 400 | 500 | 600 |
|---|---|---|---|---|---|
| RJSD-FF | $0.374 \pm 0.100$ | $0.811 \pm 0.012$ | $0.996 \pm 0.001$ | $\mathbf{1.000 \pm 0.000}$ | $\mathbf{1.000 \pm 0.000}$ |
| RJSD-RFF | $0.184 \pm 0.025$ | $0.320 \pm 0.029$ | $0.436 \pm 0.030$ | $0.644 \pm 0.037$ | $0.800 \pm 0.051$ |
| RJSD-D | $0.352 \pm 0.084$ | $\mathbf{0.898 \pm 0.108}$ | $\mathbf{1.000 \pm 0.000}$ | $\mathbf{1.000 \pm 0.000}$ | $\mathbf{1.000 \pm 0.000}$ |
| MMD-O | $0.148 \pm 0.035$ | $0.221 \pm 0.042$ | $0.283 \pm 0.042$ | $0.398 \pm 0.050$ | $0.498 \pm 0.035$ |
| MMD-D | $\mathbf{0.449 \pm 0.124}$ | $0.704 \pm 0.182$ | $0.985 \pm 0.010$ | $0.999 \pm 0.003$ | $\mathbf{1.000 \pm 0.000}$ |
| C2ST-L | $0.254 \pm 0.126$ | $0.424 \pm 0.113$ | $0.818 \pm 0.102$ | $0.967 \pm 0.029$ | $0.994 \pm 0.010$ |
| C2ST-S | $0.181 \pm 0.112$ | $0.364 \pm 0.104$ | $0.759 \pm 0.121$ | $0.945 \pm 0.042$ | $0.986 \pm 0.014$ |

covariances. Here, we perform two-sample testing increasing the number of samples per blob ($N = 9 \times$ samples per blob). Fig 4(a) presents the results. We can clearly see that RJSD-FF, RJSD-D, and JSD outperform all SOTA methods. We can conclude that even for a small number of samples the RJSD-based methods exhibit high test power.

**High-Dimensional Gaussian Mixtures (Liu et al., 2020):** In this dataset, $\mathbb{P}$ and $\mathbb{Q}$ have the same modes, and their covariances differ only on a single dimension. See Liu et al. (2020) for details. We test both, changing the number of samples while keeping the dimension constant ($d = 10$) and maintaining the number of samples ($N = 4000$) while modifying the dimensionality. Figs. 4(b) and 4(c) display the results. RJSD-D and RJSD-FF are the winners in most settings, although C2ST-L performs better at higher dimensions.

**Higgs dataset (Baldi et al., 2014):** Following Liu et al. (2020) we perform two-sample testing on the Higgs dataset ($d = 4$) as we increase the number of samples. Fig. 4(d) shows the results. Once again, RJSD-D and RJSD-FF outperform the baselines in almost all scenarios.

**MNIST generative model:** Here, we train RJSD models to distinguish between the distribution $\mathbb{P}$ of MNIST digits and the distribution $\mathbb{Q}$ of generated samples from a pretrained deep convolutional generative adversarial network (DCGAN) (Radford et al., 2015). Table 3 reports the average test power for all methods as we increase the number of samples. RJSD-D consistently outperforms the compared methods, except with the lowest number of observations.

## 6 CONCLUSIONS

We introduce the representation Jensen-Shannon divergence (RJSD), a novel measure based on embedding distributions in a feature space allowing the construction of non-parametric estimators based on Fourier features. Notably, this estimator demonstrates scalability, differentiability, making it suitable for diverse machine-learning problems. We demonstrate that RJSD provides a lower bound on the classical Jensen-Shannon divergence leading to a variational estimator of high precision compared to related approaches. We leverage this novel divergence to train generative networks, and the empirical results show that RJSD effectively mitigates mode collapse yielding generative models that produce more accurate and diverse results. Furthermore, when applied to two-sample testing, RJSD surpasses other SOTA techniques demonstrating superior performance and reliability to discriminate between distributions. These findings highlight the significant practical implications of our divergence measure.

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
