# THE REPRESENTATION JENSEN-SHANNON DIVERGENCE (SUPPLEMENTARY MATERIAL)

Next, we present the proofs of the theorems and properties described in the paper. Additionally, we include implementation details for all the experiments reported. All the codes to reproduce the results in the paper can be found in the following anonymous Github repository: `https://anonymous.4open.science/r/representationJSD-0705`.

## A    PROOFS OF THEOREMS

### A.1    PROOF THEOREM 1

*Proof.*

$$H(p,\hat{p}) = -\int_{\mathcal{X}} p(x) \log \hat{p}(x) dx$$

$$= -\int_{\mathcal{X}} \log\left[\frac{1}{h} \langle \phi(x), C_{\mathbb{P}}\phi(x)\rangle\right] p(x) dx$$

$$= -\int_{\mathcal{X}} \log\left[\frac{1}{h} \langle \phi(x), C_{\mathbb{P}}\phi(x)\rangle\right] d\mathbb{P}(x)$$

$$= -\int_{\mathcal{X}} \sum_{i=1}^{N_{\mathcal{H}}} \langle \phi(x), u_i\rangle \log(\lambda_i) \langle u_i, \phi(x)\rangle d\mathbb{P}(x) + \int_{\mathcal{X}} \log(h) d\mathbb{P}(x)$$

$$= -\int_{\mathcal{X}} \sum_{i=1}^{N_{\mathcal{H}}} \log(\lambda_i) \langle \phi(x)|u_i\rangle \langle u_i|\phi(x)\rangle d\mathbb{P}(x) + \log(h).$$

Notice that $\langle \phi(x), u_i\rangle \langle u_i, \phi(x)\rangle$ corresponds to the squared of the inner product between $\phi(x)$ and $u_i$, where the $u_i$'s are the eigenfunctions of $C_{\mathbb{P}}$. Therefore, the quantity $-\log(\lambda_i) \langle \phi(x), u_i\rangle \langle u_i, \phi(x)\rangle$ is always positive. Thus, we can apply Tonelli's theorem (DiBenedetto & Debenedetto, 2002) to take the sum out of the integral:

$$H(\mathbb{P}, \mathbb{P}_\phi) = -\sum_{i=1}^{N_{\mathcal{H}}} \log(\lambda_i) \int_{\mathcal{X}} \langle \phi(x), u_i\rangle \langle u_i, \phi(x)\rangle d\mathbb{P}(x) + \log(h)$$

$$= -\sum_{i=1}^{N_{\mathcal{H}}} \log(\lambda_i) \int_{\mathcal{X}} \langle u_i, \phi(x)\rangle \langle \phi(x), u_i\rangle d\mathbb{P}(x) + \log(h)$$

$$= -\sum_{i=1}^{N_{\mathcal{H}}} \log(\lambda_i) \langle u_i, C_{\mathbb{P}} u_i\rangle + \log(h)$$

$$= -\sum_{i=1}^{N_{\mathcal{H}}} \lambda_i \log \lambda_i + \log(h)$$

$$= S(C_{\mathbb{P}}) + \log(h)$$

$\square$

### A.2    PROOF THEOREM 2

We show in Theorem 1, that the covariance operator entropy estimator is equivalent to a plug-in Parzen density estimator with convergence and consistency extensively studied. In particular, Dmitriev &

Tarasenko (1974) show that there exists constants $C_1$ and $C_2$ such that

$$\Pr\{|H(\mathbb{P}) - (S(\mathbf{K_X}) + h)| < \epsilon\} > 1 - P_1(N; \epsilon), \tag{1}$$

where $P_1(N; \epsilon) = C_1 \exp\left(-\frac{C_2}{16}\epsilon^2 N^{\frac{1}{10}}\right)$.

Notice, that in the limit when $\gamma_N \to \infty$, $\kappa(x, x') = 0$ for all $x \neq x'$, and therefore, $h \to 1$, and since the series $\sum_{N=1}^{\infty} \exp(-CN^{\frac{1}{10}})$ converges, this leads to the desired result.

A.3  PROOF THEOREM 3

We approach the convergence problem from the perspective of density estimation. We start by proving the following lemma:

**Lemma 1.** *Let $\hat{p}_\gamma(x)$ be the empirical kernel density function by a Gaussian kernel with scale parameter $\gamma$. Similarly, let $\hat{p}_\omega(x)$ be the kernel density function by a Fourier feature mapping approximating the Gaussian kernel with scale parameter $\frac{\gamma}{2}$, then*

$$\sup_{x \in \mathcal{X}} |\hat{p}_\gamma(x) - \hat{p}_\omega(x)| \leq \epsilon, \tag{2}$$

*with probability at least $1 - 2^8 \left(\frac{\sqrt{d\gamma}h_\gamma \operatorname{diam}(\mathcal{X})}{2\epsilon}\right)^2 \exp\left(\frac{-D\epsilon^2}{h_\gamma^2(d+2)}\right)$*

*Proof.* We follow a similar proof as González et al. (2022) with some revised results.

$$\hat{p}_\gamma(x) = \frac{1}{h_\gamma N} \sum_{i=1}^{N} \left(\exp\left(-\frac{\gamma\|x - x_i\|^2}{2}\right)\right)^2 = \frac{1}{h_\gamma N} \sum_{i=1}^{N} \exp\left(-\gamma\|x - x_i\|^2\right)$$

Similarly,

$$\hat{p}_\omega(x) = \frac{1}{h_\gamma} \phi_\omega(x) \boldsymbol{C}_\mathbf{X} \phi_\omega^\top(x) = \frac{1}{h_\gamma} \phi_\omega(x) \left(\frac{1}{N} \sum_{i=1}^{N} \phi_\omega(x_i)^\top \phi_\omega(x_i)\right) \phi_\omega^\top(x)$$

$$= \frac{1}{h_\gamma N} \sum_{i=1}^{N} \phi_\omega(x) \phi_\omega(x_i)^\top \phi_\omega(x_i) \phi_\omega^\top(x) = \frac{1}{h_\gamma N} \sum_{i=1}^{N} \left(\phi_\omega(x) \phi_\omega(x_i)^\top\right)^2$$

Now, by Claim 1 by Rahimi & Recht (2007) we have that for a Gaussian kernel with parameter $\frac{\gamma}{2}$:

$$\sup_{x, x' \in \mathcal{X}} \left|\phi_\omega(x)\phi_\omega(x')^\top - \exp\left(-\frac{\gamma\|x - x'\|^2}{2}\right)\right| \leq \epsilon,$$

with probability at least $1 - 2^8 \left(\frac{\sqrt{d\gamma}\operatorname{diam}(\mathcal{X})}{\epsilon}\right)^2 \exp\left(\frac{-D\epsilon^2}{4(d+2)}\right) = 1 - B$.

Next, we use this result to bound the difference between the probability distributions.

$$\sup_{x \in \mathcal{X}} |\hat{p}_\gamma(x) - \hat{p}_\omega(x)| = \sup_{x \in \mathcal{X}} \left|\frac{1}{h_\gamma N} \sum_{i=1}^{N} \exp\left(-\gamma\|x - x_i\|^2\right) - \left(\phi_\omega(x)\phi_\omega(x_i)^\top\right)^2\right|$$

$$\leq \frac{1}{h_\gamma N} \sum_{i=1}^{N} \sup_{x \in \mathcal{X}} \left|\exp\left(-\gamma\|x - x_i\|^2\right) - \left(\phi_\omega(x)\phi_\omega(x_i)^\top\right)^2\right|$$

Notice that

$$\left|\exp\left(-\gamma\|x-x_i\|^2\right)-\left(\phi_\omega(x)\phi_\omega(x_i)^\top\right)^2\right|\le$$

$$\left|\exp\left(-\frac{\gamma\|x-x_i\|^2}{2}\right)-\phi_\omega(x)\phi_\omega(x_i)^\top\right|\left|\exp\left(-\frac{\gamma\|x-x_i\|^2}{2}\right)+\phi_\omega(x)\phi_\omega(x_i)^\top\right|$$

$$\le 2\left|\exp\left(-\frac{\gamma\|x-x_i\|^2}{2}\right)-\phi_\omega(x)\phi_\omega(x_i)^\top\right|$$

Therefore,

$$\sup_{x\in\mathcal{X}}|\hat{p}_\gamma(x)-\hat{p}_\omega(x)|\le\frac{2}{h_\gamma N}\sum_{i=1}^N\sup_{x\in\mathcal{X}}\left|\exp\left(-\frac{\gamma\|x-x_i\|^2}{2}\right)-\phi_\omega(x)\phi_\omega(x_i)^\top\right|$$

$$\le\frac{2}{h_\gamma N}\sum_{i=1}^N\epsilon=\frac{2\epsilon}{h_\gamma},$$

with probability at least $1-B$. Redefining $\epsilon=\frac{2\epsilon}{h_\gamma}$ and plugging this value in the probability we obtain the desired result. $\qquad\square$

Next, we use Schuster's Lemma (Dmitriev & Tarasenko, 1974) which proves that a Gaussian kernel density estimator can approximate any bounded distribution $p(x)$, and that there exists constants $C_1$ and $C_2$ such that

$$\sup_{x\in\mathcal{X}}|\hat{p}_\gamma(x)-p(x)|<\epsilon,\tag{3}$$

with probability at least $1-C_1\exp\left(\frac{-C_2 N\epsilon^2}{2\gamma}\right)$, where $\gamma=\frac{1}{2\sigma_N^2}$ and $\sigma_N=\mathcal{O}(\epsilon)$ with $\sigma_N\to 0$ as $N\to\infty$.

Combining Eqns. 2 and 3, using $\frac{\epsilon}{2}$, we obtain by applying triangle inequality that:

$$\sup_{x\in\mathcal{X}}|\hat{p}_\omega(x)-p(x)|<|\hat{p}_\omega(x)-\hat{p}_\gamma(x)|+|\hat{p}_\gamma(x)-p(x)|\epsilon<\frac{\epsilon}{2}+\frac{\epsilon}{2}=\epsilon,$$

with probability $1-\max\{P_3(N;\epsilon),P_2(D;\epsilon)\}$ where $P_3(N;\epsilon)=1-C_1\exp\left(-\frac{C_2 N\epsilon^2}{4\gamma}\right)$, and $P_2(D;\epsilon)=2^8\left(\frac{\sqrt{d\gamma}h_\gamma\,\mathrm{diam}(\mathcal{X})}{\epsilon}\right)^2\exp\left(\frac{-D\epsilon^2}{h_\gamma^2(d+2)}\right)$.

Finally, we can adapt Theorem 2 by Dmitriev & Tarasenko (1974) to show the convergence of the entropy induced by the Fourier features as follows:

$$\Pr\{|H(p)-H(\hat{p}_\omega)|<\epsilon\}>1-\max\{P_1(N;\epsilon),P_2(D;\epsilon)\},\tag{4}$$

where $P_1(N;\epsilon)=C_1\exp\left(-\frac{C_2}{16}\epsilon^2 N^{\frac{1}{10}}\right)$, and $P_2(D;\epsilon)=2^8\left(\frac{\sqrt{d\gamma}h_\gamma\,\mathrm{diam}(\mathcal{X})}{\epsilon}\right)^2\exp\left(\frac{-D\epsilon^2}{h_\gamma^2(d+2)}\right)$.

Following a similar strategy to the proof of Theorem 1, we can conclude that $H(\hat{p}_\omega)=S(C_\mathbf{X})+\log(h_\gamma)$. Finally, since both series $\sum_{N=1}^\infty\exp(-CN^{\frac{1}{10}})$ and $\sum_{D=1}^\infty\exp(-CD)$ the theorem holds. $\qquad\square$

## A.4 PROOF THEOREM 4

*Proof.* For equation 10 we have the following

$$
\begin{aligned}
D_{JS}^{\phi}(C_{\mathbb{P}}, C_{\mathbb{Q}}) =& \frac{1}{2} D_{KL}^{\phi}(C_{\mathbb{P}}, C_{\mathbb{M}}) + \frac{1}{2} D_{KL}^{\phi}(C_{\mathbb{Q}}, C_{\mathbb{M}}) \\
=& \frac{1}{2} D_{KL}^{\phi} \left( \int_{\mathcal{X}} \phi(x) \otimes \phi(x) d\mathbb{P}(x), \int_{\mathcal{X}} \phi(x) \otimes \phi(x) d\mathbb{M}(x) \right) + \\
& + \frac{1}{2} D_{KL}^{\phi} \left( \int_{\mathcal{X}} \phi(x) \otimes \phi(x) d\mathbb{Q}(x), \int_{\mathcal{X}} \phi(x) \otimes \phi(x) d\mathbb{M}(x) \right) \\
=& \frac{1}{2} D_{KL}^{\phi} \left( \int_{\mathcal{X}} \phi(x) \otimes \phi(x) d\mathbb{P}(x), \int_{\mathcal{X}} \frac{d\mathbb{M}}{d\mathbb{P}}(x) \phi(x) \otimes \phi(x) d\mathbb{P}(x) \right) + \\
& + \frac{1}{2} D_{KL}^{\phi} \left( \int_{\mathcal{X}} \phi(x) \otimes \phi(x) d\mathbb{Q}(x), \int_{\mathcal{X}} \frac{d\mathbb{M}}{d\mathbb{Q}}(x) \phi(x) \otimes \phi(x) d\mathbb{P}(x) \right).
\end{aligned}
$$

Since $D_{KL}^{\phi}$ is jointly convex (Bach, 2022), then

$$
\begin{aligned}
D_{JS}^{\phi}(C_{\mathbb{P}}, C_{\mathbb{Q}}) \leq& \frac{1}{2} \int_{\mathcal{X}} D_{KL}^{\phi} \left( \phi(x) \otimes \phi(x), \frac{d\mathbb{M}}{d\mathbb{P}}(x) \phi(x) \otimes \phi(x) \right) d\mathbb{P}(x) + \\
& + \frac{1}{2} \int_{\mathcal{X}} D_{KL}^{\phi} \left( \phi(x) \otimes \phi(x), \frac{d\mathbb{M}}{d\mathbb{Q}}(x) \phi(x) \otimes \phi(x) \right) d\mathbb{Q}(x).
\end{aligned}
$$

Notice that $\phi(x) \otimes \phi(x)$ is a rank-1 covariance operator with one eigenvalue equal $\|\phi(x)\|^2 = 1$ and one eigen vector $\phi(x)$, therefore, it can be simplified as:

$$
\begin{aligned}
D_{JS}^{\phi}(C_{\mathbb{P}}, C_{\mathbb{Q}}) \leq& \frac{1}{2} \int_{\mathcal{X}} D_{KL}^{\phi} \left( 1, \frac{d\mathbb{M}}{d\mathbb{P}}(x) \right) d\mathbb{P}(x) + \frac{1}{2} \int_{\mathcal{X}} D_{KL}^{\phi} \left( 1, \frac{d\mathbb{M}}{d\mathbb{Q}}(x) \right) d\mathbb{Q}(x) \\
=& \frac{1}{2} \int_{\mathcal{X}} D_{KL}^{\phi} \left( 1, \frac{d\mathbb{M}}{d\mathbb{P}}(x) \right) d\mathbb{P}(x) + \frac{1}{2} \int_{\mathcal{X}} D_{KL}^{\phi} \left( 1, \frac{d\mathbb{M}}{d\mathbb{Q}}(x) \right) d\mathbb{Q}(x) \\
=& \frac{1}{2} \int_{\mathcal{X}} -\log \left( \frac{d\mathbb{M}}{d\mathbb{P}}(x) \right) d\mathbb{P}(x) + \frac{1}{2} \int_{\mathcal{X}} -\log \left( \frac{d\mathbb{M}}{d\mathbb{Q}}(x) \right) d\mathbb{Q}(x) \\
=& \frac{1}{2} D_{KL}(\mathbb{P}, \mathbb{M}) + \frac{1}{2} D_{KL}(\mathbb{Q}, \mathbb{M}) = D_{JS}(\mathbb{P}, \mathbb{Q})
\end{aligned}
$$

$\square$

## A.5 PROOF THEOREM 5

*Proof.* Notice that for Gaussian Kernels, the normalizing constant only depends on the scale parameter $\gamma$ which is independent of the data, thus $h_{\mathbb{P}} = h_{\mathbb{Q}}$. Hence,

$$
\begin{aligned}
H \left( \frac{\mathbb{P} + \mathbb{Q}}{2} \right) =& -\int_{\mathcal{X}} \log \left[ \frac{1}{2h} \langle \phi^*(x), C_{\mathbb{P}} \phi^*(x) \rangle + \frac{1}{2h} \langle \phi^*(x), C_{\mathbb{Q}} \phi^*(x) \rangle \right] d\mathbb{M}(x) \\
=& -\int_{\mathcal{X}} \log \left[ \left\langle \phi^*(x), \left( \frac{C_{\mathbb{P}} + C_{\mathbb{Q}}}{2} \right) \phi^*(x) \right\rangle \right] d\mathbb{M}(x) + \log(h) \\
=& S \left( \frac{C_{\mathbb{P}} + C_{\mathbb{Q}}}{2} \right) + \log(h)
\end{aligned}
$$

where $\mathbb{M}(x) = \frac{\mathbb{P}(x)+\mathbb{Q}(x)}{2}$. Using the results from theorem 1, it is straightforward to show that

$$
\begin{aligned}
D_{JS}(\mathbb{P}, \mathbb{Q}) &= S\left(\frac{C_\mathbb{P} + C_\mathbb{Q}}{2}\right) - \frac{1}{2}(S(C_\mathbb{P}) + S(C_\mathbb{Q})) + \log(h) - \frac{1}{2}(\log(h) + \log(h)) \\
&= S\left(\frac{C_\mathbb{P} + C_\mathbb{Q}}{2}\right) - \frac{1}{2}(S(C_\mathbb{P}) + S(C_\mathbb{Q})) \\
&= D_{JS}^{\phi^*}(C_\mathbb{P}, C_\mathbb{Q}).
\end{aligned}
$$

$\square$

### A.6 PROOF THEOREM 6

*Proof.* To prove this theorem, we use (Proposition 4.e) by Bach (2022). We have that

$$
D_{KL}^\phi(C_\mathbb{P}|C_\mathbb{Q}) \geq \frac{1}{2}\|C_\mathbb{P} - C_\mathbb{Q}\|_*^2 \geq \frac{1}{2}\|C_\mathbb{P} - C_\mathbb{Q}\|_{HS}^2,
$$

where $\|\cdot\|_*$ and $\|\cdot\|_{HS}$ denote the nuclear norm and the Hilbert-Schmidt norm respectively. Since

$$
\begin{aligned}
D_{JS}^\phi(C_\mathbb{P}, C_\mathbb{Q}) &= \frac{1}{2}D_{KL}^\phi(C_\mathbb{P}, C_\mathbb{M}) + \frac{1}{2}D_{KL}^\phi(C_\mathbb{P}, C_\mathbb{M}) \\
&\geq \frac{1}{4}\left\|C_\mathbb{P} - \frac{1}{2}(C_\mathbb{P} + C_\mathbb{Q})\right\|_*^2 + \frac{1}{4}\left\|C_\mathbb{Q} - \frac{1}{2}(C_\mathbb{P} + C_\mathbb{Q})\right\|_*^2 \\
&\geq \frac{1}{4}\left\|\frac{1}{2}C_\mathbb{P} - \frac{1}{2}C_\mathbb{Q}\right\|_*^2 + \frac{1}{4}\left\|\frac{1}{2}C_\mathbb{Q} - \frac{1}{2}C_\mathbb{P}\right\|_*^2 = \frac{1}{8}\|C_\mathbb{P} - C_\mathbb{Q}\|_*^2
\end{aligned}
$$

and thus, $D_{JS}^\phi(C_\mathbb{P}, C_\mathbb{Q}) \geq \frac{1}{8}\|C_\mathbb{P} - C_\mathbb{Q}\|_*^2 \geq \frac{1}{8}\|C_\mathbb{P} - C_\mathbb{Q}\|_{HS}^2$.

Now, let $\phi : \mathcal{X} \mapsto \mathcal{H}$ then, and $\{e_\alpha\}$ be an orthonormal basis in $\mathcal{H}$, we have that

$$
\begin{aligned}
\operatorname{Tr}\left(\phi(x) \otimes \phi(x)\phi(y) \otimes \phi(y)\right) &= \sum_\alpha \langle \phi(x) \otimes \phi(x)\phi(y) \otimes \phi(y)e_\alpha, e_\alpha \rangle \\
&= \sum_\alpha \langle \phi(x)\langle\phi(x), \phi(y) \otimes \phi(y)e_\alpha\rangle, e_\alpha \rangle \\
&= \sum_\alpha \langle \phi(x)\langle\phi(x), \phi(y)\langle\phi(y), e_\alpha\rangle\rangle, e_\alpha \rangle \\
&= \sum_\alpha \langle \phi(x)\langle\phi(x), \phi(y)\rangle\langle\phi(y), e_\alpha\rangle, e_\alpha \rangle \\
&= \sum_\alpha \langle \phi(x), e_\alpha\rangle\langle\phi(x), \phi(y)\rangle\langle\phi(y), e_\alpha \rangle \\
&= \langle\phi(x), \phi(y)\rangle \sum_\alpha \langle\phi(x), e_\alpha\rangle\langle\phi(y), e_\alpha\rangle = \langle\phi(x), \phi(y)\rangle\langle\phi(x), \phi(y)\rangle \\
&= \langle\phi(x), \phi(y)\rangle^2 = \kappa(x, y)^2
\end{aligned}
$$

Note that for $T : \mathcal{H} \mapsto \mathcal{H}$, $\operatorname{Tr}(T^*T) = \sum_\alpha \langle Te_\alpha, Te_\alpha \rangle = \|T\|_{HS}^2$. In particular, if we have that $T = \phi(x) \otimes \phi(x) - \phi(y) \otimes \phi(y)$,

$$
\begin{aligned}
\|\phi(x) \otimes \phi(x) - \phi(y) \otimes \phi(y)\|_{HS}^2 &= \operatorname{Tr}(\phi(x) \otimes \phi(x)\phi(x) \otimes \phi(x)) - 2\operatorname{Tr}(\phi(x) \otimes \phi(x)\phi(y) \otimes \phi(y)) \\
&\quad + \operatorname{Tr}(\phi(y) \otimes \phi(y)\phi(y) \otimes \phi(y)) \\
&= \kappa^2(x, x) - 2\kappa^2(x, y) + \kappa^2(y, y)
\end{aligned}
$$

Finally, note that

$$
\begin{aligned}
\|C_\mathbb{P} - C_\mathbb{Q}\|_{HS}^2 &= \operatorname{Tr}(\mathbb{E}_\mathbb{P}[\phi(x) \otimes \phi(x)]\mathbb{E}_{\mathbb{P}'}[\phi(x) \otimes \phi(x)]) - 2\operatorname{Tr}(\mathbb{E}_\mathbb{P}[\phi(x) \otimes \phi(x)]\mathbb{E}_\mathbb{Q}[\phi(y) \otimes \phi(y)]) \\
&\quad + \operatorname{Tr}(\mathbb{E}_\mathbb{Q}[\phi(y) \otimes \phi(y)]\mathbb{E}_{\mathbb{Q}'}[\phi(y) \otimes \phi(y)]) \\
&= \operatorname{Tr}(\mathbb{E}_{\mathbb{P},\mathbb{P}'}[\phi(x) \otimes \phi(x)\phi(x') \otimes \phi(x')]) - 2\operatorname{Tr}(\mathbb{E}_{\mathbb{P},\mathbb{Q}}[\phi(x) \otimes \phi(x)\phi(y) \otimes \phi(y)]) \\
&\quad + \operatorname{Tr}(\mathbb{E}_{\mathbb{Q},\mathbb{Q}'}[\phi(y) \otimes \phi(y)\phi(y') \otimes \phi(y')]) \\
&= \mathbb{E}_{\mathbb{P},\mathbb{P}'}[\kappa^2(x, x')] - 2\mathbb{E}_{\mathbb{P},\mathbb{Q}}[\kappa^2(x, y)] + \mathbb{E}_{\mathbb{Q},\mathbb{Q}'}[\kappa^2(y, y')],
\end{aligned}
$$

which corresponds to MMD with kernel $\kappa^2(\cdot, \cdot)$. $\square$

## A.7 PROOF LEMMA 1

*Proof.* Notice that the sum of covariance matrices in the RKHS corresponds to the concatenation of samples in the input space, that is:

$$
\begin{aligned}
\pi_1 \boldsymbol{C}_\mathbf{X} + \pi_2 \boldsymbol{C}_\mathbf{Y} &= \frac{1}{N+M} \boldsymbol{\Phi}_X^\top \boldsymbol{\Phi}_X + \frac{1}{N+M} \boldsymbol{\Phi}_Y^\top \boldsymbol{\Phi}_Y \\
&= \frac{1}{N+M} \begin{bmatrix} \boldsymbol{\Phi}_X^\top & \boldsymbol{\Phi}_Y^\top \end{bmatrix} \begin{bmatrix} \boldsymbol{\Phi}_X \\ \boldsymbol{\Phi}_Y \end{bmatrix} \\
&= \frac{1}{N+M} \boldsymbol{\Phi}_Z^\top \boldsymbol{\Phi}_Z = \mathbf{C}_Z,
\end{aligned}
$$

where $\boldsymbol{\Phi}_Z = \begin{bmatrix} \boldsymbol{\Phi}_X^\top & \boldsymbol{\Phi}_Y^\top \end{bmatrix}^\top \in \mathbb{R}^{(N+M) \times D}$ contains the mappings of the mixture (concatenation) of the samples in the input space $\mathbf{Z}$. Since the spectrum of $\boldsymbol{C}_\mathbf{Z}$ and $\mathbf{K}_\mathbf{Z}$ have the same non-zero eigenvalues, $S(\pi_1 \boldsymbol{C}_\mathbf{X} + \pi_2 \boldsymbol{C}_\mathbf{Y}) = S(\boldsymbol{C}_\mathbf{Z}) = S(\mathbf{K}_\mathbf{Z})$ □

## A.8 CONVERGENCE OF RJSD KERNEL-BASED ESTIMATOR

Bach (2022) shows the following result regarding the convergence of the empirical estimator $S(\mathbf{K}_\mathbf{X})$ to $S(C_\mathbb{P})$.

**Proposition 1.** *(Bach, 2022)[Proposition 7] Assume that $\kappa$ is a continuous positive definite kernel on the compact set $\mathcal{X}$, with $\kappa(x, x) = 1$ for all $x \in \mathcal{X}$. Also, assume that $\mathbb{P}$ has a density with respect to the uniform measure which is greater than $\alpha < 1$. Finally, assume that $c = \int_0^\infty \sup_{x \in \mathcal{X}} \langle \phi(x), (C_\mathbb{P} + \lambda I)^{-1} \phi(x) \rangle^2 d\lambda$ is finite. Given i.i.d. samples $\boldsymbol{X} = \{\boldsymbol{x}_i\}_{i=1}^N$, then:*

$$
\mathbb{E}\left[S(\mathbf{K}_\mathbf{X}) - S(C_\mathbb{P})\right] \leq \frac{1 + c(8\log(N))^2}{\alpha N} + \frac{17}{\sqrt{N}}(2\sqrt{c} + \log(N)). \tag{5}
$$

Since RJSD corresponds to the empirical estimation of three different covariance operator entropies, and assuming $N = M$ for simplicity, it is easy to show that:

$$
\mathbb{E}\left[D_{JS}^\kappa(\mathbf{X}, \mathbf{Y}) - D_{JS}^\phi(C_\mathbb{P}, C_\mathbb{Q})\right] \leq 3\left[\frac{1 + c(8\log(N))^2}{\alpha N} + \frac{17}{\sqrt{N}}(2\sqrt{c} + \log(N))\right]. \tag{6}
$$

Therefore, we can conclude that $D_{JS}^\kappa(\mathbf{X}, \mathbf{Y})$ converges to the population quantity $D_{JS}^\phi(C_\mathbb{P}, C_\mathbb{Q})$ at a rate $\mathcal{O}\left(\frac{1}{\sqrt{N}}\right)$.

## B ALGORITHMS

Algorithms 1, 2, and 3 describe the procedure to estimate JS divergence regularly and with Exponential Moving Averages (EMA), and to train GANs based on representation JS divergence.

## C EXPERIMENTS IMPLEMENTATION DETAILS

### C.1 NEURAL JS DIVERGENCE ESTIMATION

**Jensen-Shannon divergence between Cauchy distributions:** The Jensen-Shannon for two Cauchy distributions $\mathbb{P} \sim p(x; l_p, s_p)$ and $\mathbb{Q} \sim p(x; l_q, s_q)$ can be estimated as (Nielsen & Okamura, 2022):

$$
D_{JS}(\mathbb{P}, \mathbb{Q}) = \log\left(\frac{2\sqrt{(l_p - l_q)^2 + (s_p + s_q)^2}}{\sqrt{(l_p - l_q)^2 + (s_p + s_q)^2} + 2\sqrt{s_p s_q}}\right)
$$

In this experiment we set $s_p = s_q = 1$, and $l_p = 0$. Then we calculate the value of $l_q$ to achieve a specified divergence value in the set $\log(2) \times \{0.2, 0.4, 0.6, 0.8, 0.99\}$.

---

**Algorithm 1** JS divergence estimation

---

**Input:** $\mathbf{X} \sim \mathbb{P}, \mathbf{Y} \sim \mathbb{Q}, \eta$
1: $\omega \leftarrow$ Initialize network parameters parameters.
2: **for** $T = 1$ : Number of epochs **do**
3:      $\boldsymbol{\Phi}_X \leftarrow \phi_\omega \circ f_\omega(\mathbf{X})$
4:      $\boldsymbol{\Phi}_Y \leftarrow \phi_\omega \circ f_\omega(\mathbf{Y})$
5:      $\boldsymbol{C}_{\mathbf{X}} \leftarrow \frac{1}{N} \boldsymbol{\Phi}_X^\top \boldsymbol{\Phi}_X$
6:      $\boldsymbol{C}_{\mathbf{Y}} \leftarrow \frac{1}{M} \boldsymbol{\Phi}_Y^\top \boldsymbol{\Phi}_Y$
7:      $D_{JS}^\omega(\mathbf{X}, \mathbf{Y}) = S\left(\pi_1 \boldsymbol{C}_{\mathbf{X}} + \pi_2 \boldsymbol{C}_{\mathbf{Y}}\right) - \left(\pi_1 S(\boldsymbol{C}_{\mathbf{X}}) + \pi_2 S(\boldsymbol{C}_{\mathbf{Y}})\right)$    ▷ as in Eqn. 14
8:      $\omega \leftarrow \omega + \eta \nabla_{\text{Adam}} D_{JS}^\omega(\mathbf{X}, \mathbf{Y})$          ▷ Maximize the divergence
9: **end for**
**Output:** $\widehat{D_{JS}}(\mathbb{P}, \mathbb{Q}) = D_{JS}^\omega(\mathbf{X}, \mathbf{Y})$

---

**Algorithm 2** JS divergence estimation EMA

---

**Input:** $\mathbf{X} \sim \mathbb{P}, \mathbf{Y} \sim \mathbb{Q}, \eta, \alpha$
1: $\omega \leftarrow$ Initialize network parameters parameters.
2: **for** $T = 1$ : Number of epochs **do**
3:      $\boldsymbol{\Phi}_X; \boldsymbol{\Phi}_Y$                                    ▷ Compute the mappings
4:      $\boldsymbol{C}_{\mathbf{X}}; \boldsymbol{C}_{\mathbf{Y}}$                       ▷ Compute the covariance matrices
5:      **if** $T = 1$ **then**
6:          $\widehat{\mathbf{C}}_X[T] = \boldsymbol{C}_{\mathbf{X}}$
7:          $\widehat{\mathbf{C}}_Y[T] = \boldsymbol{C}_{\mathbf{Y}}$             ▷ Store previous covariance matrices
8:      **else**
9:          $\widehat{\mathbf{C}}_X[T] \leftarrow (1 - \alpha)\widehat{\mathbf{C}}_X[T - 1] + \alpha \boldsymbol{C}_{\mathbf{X}}$
10:        $\widehat{\mathbf{C}}_Y[T] \leftarrow (1 - \alpha)\widehat{\mathbf{C}}_Y[T - 1] + \alpha \boldsymbol{C}_{\mathbf{Y}}$ ▷ Compute EMA covariance matrices
11:      **end if**
12:      $D_{JS}^\omega(\mathbf{X}, \mathbf{Y}) = S\left(\pi_1 \widehat{\mathbf{C}}_X[T] + \pi_2 \widehat{\mathbf{C}}_Y[T]\right) - \left(\pi_1 S(\widehat{\mathbf{C}}_X[T]) + \pi_2 S(\widehat{\mathbf{C}}_Y[T])\right)$
13:      $e\omega \leftarrow e\omega + \eta \nabla_{\text{Adam}} D_{JS}^\omega(\mathbf{X}, \mathbf{Y})$        ▷ Maximize the divergence
14: **end for**
**Output:** $\widehat{D_{JS}}(\mathbb{P}, \mathbb{Q}) = D_{JS}^\omega(\mathbf{X}, \mathbf{Y})$

---

**Algorithm 3** Representation JS divergence GAN

---

**Input:** $\mathbf{X}_P = \{\mathbf{X}_i\}_{i=1}^k \sim \mathbb{P}$
1: $\theta, \omega \leftarrow$ Initialize network parameters parameters.
2: **for** $T = 1$ : Number of epochs **do**
3:      **for** $i = 1 : k$ **do**
4:          $\mathbf{Y}_i^\theta = g_\theta(\mathbf{z})$          ▷ Generated batch from random noise $\mathbf{z}$
5:          $\omega \leftarrow \omega + \eta_d \nabla_{\text{Adam}} D_{JS}^\omega(\mathbf{X}_i, \mathbf{Y}_i^\theta)$      ▷ Maximize the divergence
6:          $\boldsymbol{\theta} \leftarrow \boldsymbol{\theta} - \eta_g \nabla_{\text{Adam}} D_{JS}^\omega(\mathbf{X}_i, \mathbf{Y}_i^\theta)$      ▷ Minimize the divergence
7:      **end for**
8: **end for**

---

**Implementation details and hyperparameters:** Next, we show all the configurations that we used to perform the JS divergence estimation using the representation JS divergence. For this experiment, we use the covariance estimator by using Random Fourier Features (RFFs) to approximate a Gaussian kernel. We chose 50 RFFs and an initial kernel length scale $\sigma = 2$. We set the learning rate as $l_r = 0.001$ and we use the default $\beta_1$ and $\beta_2$ of the Adam optimizer. We did not use a deep neural network, but the Fourier Features layer by itself. We then applied algorithm 1 to learn $\boldsymbol{\omega}$ and $\sigma$. We also implemented a version of the algorithm using exponential moving averages (EMA) of the

Table 2: Architectures mode collapse experiments

| Generator | Discriminator | DFFN |
|---|---|---|
| Linear(32,256) | Linear(2,256) | Linear(2,256) |
| Leaky ReLU(0.01) | Leaky ReLU(0.01) | Leaky ReLU(0.01) |
| Linear(256,256) | Linear(256,256) | Linear(256,256) |
| Leaky ReLU(0.01) | Leaky ReLU(0.01) | Leaky ReLU(0.01) |
| Linear(256,256) | Linear(256,256) | Linear(256,256) |
| Leaky ReLU(0.01) | Leaky ReLU(0.01) | Leaky ReLU(0.01) |
| Linear(256,256) | Linear(256,256) | Linear(256,256) |
| tanH() | Leaky ReLU(0.01) | Leaky ReLU(0.01) |
| Linear(256,2) | Linear(256, 1) | Fourier Features Layer(256, 8) |

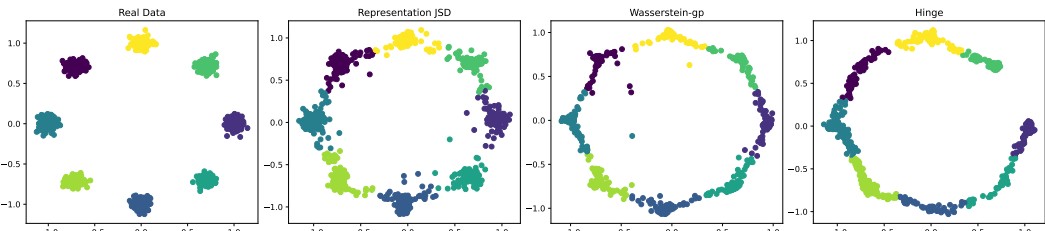

Figure 1: Evaluation of differences between the generated modes and the real modes. The generated data is clustered in 8 modes, and covariances and means are estimated. Then the KL divergence between the real and generated distributions is computed

covariance matrices. Specifically, we store information from past covariances and applied EMA to smooth out the estimation. Algorithm 2 shows a detailed explanation of this implementation.

## C.2 GENERATIVE ADVERSARIAL NETWORKS

### C.2.1 MODE COLLAPSE EXPERIMENTS

To perform the mode collapse experiments, we used the code provided in `https://github.com/ChristophReich1996/Mode_Collapse` to compare GAN losses. To make a fair comparison, the generators of the compared losses are the same. The representation JS divergence does not rely on a discriminator or classifier, however, we used a Deep Fourier Features Network (DFFN) with a similar architecture to the discriminator that we used for the compared losses. Table 2 describes in detail the architectures employed.

For the representation JS divergence, we set the learning rate for the discriminator as $l_d = 1 \times 10^{-4}$ and the learning rate of the generator as $l_g = 5 \times 10^{-4}$. For Wasserstein-GP and Hinge losses, we used $l_d = l_g = 1 \times 10^{-4}$ For the standard GAN loss we used $l_d = 5 \times 10^{-4}$ and $l_r = 1 \times 10^{-4}$ .We chose random uniform noise $z \sim \mathcal{U}^{32}[0, 1]$.

### C.2.2 STACKED MNIST IMPLEMENTATION DETAILS

Next, we describe the architecture and hyperparameters selection that we used to train a GAN in the stacked MNIST dataset, as well as some practical considerations. We used the standard DCGAN generator architecture (Radford et al., 2015) and slightly modified the discriminator architecture to incorporate a Fourier Feature Layer. Table 3 describes in detail the architecture employed. As you can see, we removed all batch norm layers in the discriminator and added two linear layers before the Fourier Feature mapping to reduce the high dimensionality of the last convolutional layer. We resized the images to $64 \times 64 \times 3$ to be compatible with the standard DCGAN architecture.

We draw $z$ from a truncated Gaussian of 100 dimensions, with truncation parameter $\tau = 0.5$, where values that fall outside $\tau$ times the standard deviation are resampled to fall inside the range. This is known as the truncation trick.

Table 3: Architecture GAN on stacked MNIST. We use the following notation for the convolutional layers: ConvLayer(input channels, output channels, kernel size, stride, padding)

| Generator | DFFN |
|---|---|
| ConvTranspose2d(100,512,4,1,0) | Conv2d(3, 64, 4, 2, 1) |
| Batchnorm() | LeakyReLU(0.2) |
| ReLU() | Conv2d(64, 128, 4, 2, 1) |
| ConvTranspose2d(512,256,4,2,1) | LeakyReLU(0.2) |
| Batchnorm() | Conv2d(128, 256, 4, 2, 1 |
| ReLU() | LeakyReLU(0.2) |
| ConvTranspose2d(256,128,4,2,1) | Conv2d(256, 512, 4, 2, 1) |
| Batchnorm() | LeakyReLU(0.2) |
| ReLU() | Flatten() |
| ConvTranspose2d(128,64 ,4,2,1) | Linear(8192, 512) |
| Batchnorm() | LeakyReLU(0.2) |
| ReLU() | Linear(512, 256) |
| ConvTranspose2d(64,3,4,2,1) | Fourier Features Layer (256, 4) |
| tanH() | |

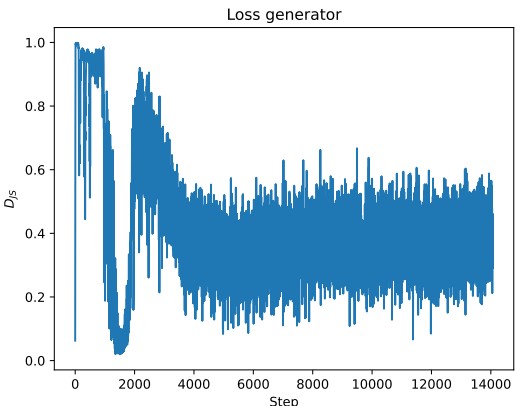

Figure 2: Loss GAN on the stacked MNIST dataset.

We set the batch size as 64, and we train the GAN for 15 epochs. We set the discriminator's learning rate as $l_d = 2.0 \times 10^{-5}$ whereas the generator's as $l_g = 1.0 \times 10^{-4}$. We chose $\beta_1 = 0.5$ and $\beta_2 = 0.999$ as the hyperparameters of the Adam optimizer.

Here are some practical considerations to train a GAN with the representation JS divergence. First, the learning rates are crucial. If we use a high learning rate for the discriminator, it leads to a 0.99 divergence value, which the generator is not able to reduce. In contrast, if the discriminator's learning rate is too small, then the divergence will remain close to 0 and the algorithm will not learn. So it is really important to choose the learning rates so that the divergence can grow quickly in the first steps but then the generator should be able to lower the divergence quickly too. Figure 2 shows the loss behavior during training with the selected learning rates. We also observed that a small number of Fourier Features (4 Fourier features, leading to 8 dimensions) lead to better results and make the algorithm easier and faster to train. We observed empirically that a large number of Fourier Features — although they could potentially capture richer information — makes the model prone to overfitting, yielding a high divergence regardless of what the generator does, the divergence stays high.

## C.3 TWO-SAMPLE TESTING

**Procedure details** The procedure is the following: for synthetic datasets, we create the sets $\mathbf{X}_{train} \in \mathbb{R}^{N \times d}$ and $\mathbf{Y}_{train} \in \mathbb{R}^{M \times d}$. Then, we learn the kernel/covariance/classifier for each of the methods on that training set. We then sample a testing set $\mathbf{X}_{test} \in \mathbb{R}^{N \times d}$ and $\mathbf{Y}_{test} \in \mathbb{R}^{M \times d}$ and perform a permutation test. We compute the statistic on the testing set and perform 100 permutations

to generate the surrogate of the distribution of the measurement under the null hypothesis. Finally, we compute the rejection threshold, and if the statistic is greater than this threshold we reject the null hypothesis. This is done for 100 independent testing sets. Finally, we repeat the experiment ten times and compute the average test power.

**RJSD-D test implementation details** In this experiment, we try a slightly different implementation of the proposed covariance estimator using a deep Fourier Features network. We explore the idea of deep kernel learning by following a similar approach to Liu et al. (2020), where a characteristic kernel $\kappa_\omega(x, y)$ is built as follows:

$$\kappa_\omega(x, y) = [(1 - \epsilon)\kappa_1(f_\omega(x), f_\omega(y)) + \epsilon]\,\kappa_2(x, y), \tag{7}$$
$$= (1 - \epsilon)\kappa_1(f_\omega(x), f_\omega(y))\kappa_2(x, y) + \epsilon\kappa_2(x, y) \tag{8}$$

where $f_\omega : \mathcal{X} \to \mathcal{F}$ is a deep network that extracts features from the data, allowing the kernel to have more flexibility to capture more accurately the structure of complicated distributions. $0 < \epsilon < 1$ and $\kappa_1$ and $\kappa_2$ are Gaussian kernels. Notice, that the kernel of the deep network features, $\kappa_1$, is multiplied by another kernel $\kappa_2$ on the input space. This approach prevents the deep kernel from considering distant points in the input space as very similar.

In this work, we extend this idea to covariance operators and we propose a similar approach to learn deep covariance operators by learning an explicit mapping to the RKHS of a deep kernel. In first place, consider the product $\kappa_p(x, y) = \kappa_1(f_\omega(x), f_\omega(y))\kappa_2(x, y)$. Assuming $\kappa_\sigma = \kappa_1 = \kappa_2$ are Gaussian kernels with bandwidth $\sigma$, then $\kappa_p(x, y) = \kappa_\sigma(f_\omega(x) \oplus x, f_\omega(y) \oplus y)$, where $\oplus$ stands for concatenation of the dimensions, that is, $\kappa_p$ would be the kernel applied to the concatenation of the features from the deep network and the features in the input space. Afterward, we can use Fourier Features to learn an explicit mapping $\phi_\omega : \mathcal{X} \oplus \mathcal{F} \to \mathcal{H}_\phi$ to approximate a given shift-invariant kernel. Notice, that this approach is nothing but a linear layer with random weights $e\omega \sim p(e\omega)$ and sines and cosines as activation functions. Therefore $\kappa_p(x, y) \approx \langle\phi_\omega(f_\omega(x) \oplus x), \phi_\omega(f_\omega(y) \oplus y)\rangle_{\mathcal{H}_\phi}$. $\kappa_2(x, y)$ can be similarly approximated through a Fourier Feature mapping $\psi_\omega : \mathcal{X} \to \mathcal{H}_\psi$ applied directly on the samples in the input space, that is $\kappa_2(x, y) \approx \langle\psi_\omega(x), \psi_\omega(y)\rangle_{\mathcal{H}_\psi}$.

Finally, consider the whole kernel $\kappa_\omega(x, y) = (1 - \epsilon)\kappa_p(x, y) + \epsilon\kappa_2(x, y)$, which is the direct sum of two kernels with approximated explicit mappings $\phi_\omega$ and $\psi_\omega$ respectively. By the properties of RKHS, it can be shown that:

$$\kappa_\omega(x, y) = (1 - \epsilon)\kappa_p(x, y) + \epsilon\kappa_2(x, y)$$
$$\approx \langle\varphi_\omega(x), \varphi_\omega(y)\rangle_{\mathcal{H}}, \tag{9}$$

where $\varphi_\omega(x) = \left[(1 - \epsilon)^{\frac{1}{2}}\phi_\omega(x')\right] \oplus \left[\epsilon^{\frac{1}{2}}\psi_\omega(x)\right]$, $\mathcal{H} = \mathcal{H}_\psi \oplus \mathcal{H}_\phi$ and $x' = f_\omega(x) \oplus x$.

This procedure allows us to obtain an explicit mapping to the RKHS from a deep kernel that can be used to compute an explicit covariance operator. Consequently, this covariance operator can be optimized to maximize the JS divergence between the distributions. Note, that we can learn the parameters of the network $f_\omega$ as well as the Fourier Features $e\omega$ and the kernel bandwidth $\sigma$.

**Two-sample testing implementation details** We run all baselines using the official implementation, that is MMD-O and MMD-D (Liu et al., 2020)[1], C2ST-S and C2ST-L (Cheng & Cloninger, 2022) [2]. We follow all the configuration and architecture proposed by Liu et al. (2020). To perform RJSD-D, we used the same architecture as MMD-D, although, we add a Fourier Feature layer where MMD-D computes a kernel. Table 4 shows the details of the architecture used. The base network consists of five fully connected layers, and the number of neurons in hidden and output layers is set to 50 for Blob, $3 \times d$ for HDGM, and 20 for the Higgs dataset, where $d$ is the dimension of the dataset. Also, the number of Fourier Features for all JSD-based tests is set to 50 for Blob, 15 for HDGM, and 15 for the Higgs dataset.

---

[1] https://github.com/fengliu90/DK-for-TST
[2] https://github.com/xycheng/net_logit_test/tree/main

Table 4: Architecture of the Deep Fourier Features Network (DFFN) and the Deep Convolutional Fourier Features Network(DCFFN) used in two sample testing. $d$ is the input dimensionality, $H$ is the number of hidden neurons and $FF$ is the number of Fourier Features. The Convolutional Layers follow the same notation as 3

| DFFN | DCFFN (MNIST) |
|---|---|
| Linear($d$, $H$) | Conv2d(3,16,3,2,1) |
| Softplus() | LeakyReLU(0.2) |
| Linear($H$,$H$) | Conv2d(16,32,3,2,1) |
| Softplus() | LeakyReLU(0.2) |
| Linear($H$,$H$) | Conv2d(32,64,3,2,1) |
| Softplus() | LeakyReLU(0.2) |
| Linear($H$,$H$) | Conv2d(64,128,3,2,1) |
| Fourier Features Layer ($H$, $FF$) | LeakyReLU(0.2) |
| | Linear (128,512) |
| | ReLU |
| | Linear (512,100) |

**Two-sample testing implementation details on MNIST** DCFFN is the model used on MNIST and it is described in Table 4. This model is the same proposed by Liu et al. (2020) except that we remove the batch normalization layers between the convolutional layers. For RJSD-D, we set the batch size to 100 and the number of epochs to 200 for MNIST. We set the learning rate to 0.05 for MNIST. We set the number of Fourier Features to 10 for MNIST.

For RJSD-RFF, we use full batch size to train it. We set the number of epochs to 200 for MNIST. We set the learning rate to 0.01 for MNIST and 0.001. We set the number of Fourier Features to 200 for MNIST.

For RJSD-FF, we use full batch size to train it. We set the number of epochs to 200 for MNIST. We set the learning rate to 0.05 for MNIST.

We use Adam optimizer to optimize 1) The kernel length scale $\sigma$ in RJSD-K and RJSD-RFF 2) the Fourier Features $\boldsymbol{\omega}$ and kernel length scale $\sigma$ in RJSD-FF and 3) the network parameters $\omega \in \Omega$, the Fourier Features $\boldsymbol{\omega}$ and the kernel length scale $\sigma$ in RJSD-D. For the blobs dataset, we set the learning rate of RJSD-FF, RJSD-RFF and RJSD-D as $l_r = 1 \times 10^{-3}$. For the HDGM, we set the learning rate for RJSD-FF and RJSD-RFF as $l_r = 5 \times 10^{-3}$, and the one for RJSD-D as $l_r = 5 \times 10^{-2}$. In the Higgs dataset, we set all the learning rates as $l_r = 1 \times 10^{-2}$.

Figures 3 4,5,6 show the average test power and the standard deviation of each of the implemented test.

### C.4 LIMITATIONS

Although the representation Jensen-Shannon divergence shows promising results, there are still some aspects that require further research. So far, the number of Fourier Features to build the reproducing kernel Hilbert space (RKHS) has been chosen arbitrarily. Empirically, we have found that choosing $D << N$ usually leads to better results, which could seem counter-intuitive with the kernel theory that usually induces a high-dimensional space. Also, using the kernel-based estimator for maximization purposes would require enforcing constraints on the scale of the data since this estimator can potentially exhibit rank-inconsistency of the matrices, that is $\max \text{Rank}(\mathbf{K_Z}) = N + M$, $\max \text{Rank}(\mathbf{K_X}) = N$ and $\max \text{Rank}(\mathbf{K_Y}) = M$. If the data scale is not kept fixed, trivial maximization of the divergence by just spreading all the samples far apart in the space, or equivalently for a Gaussian kernel by decreasing the length-scale $\sigma$ can occur. For this reason, we employed the covariance-based estimator with a finite dimension (explicit feature space) in most of our experiments.

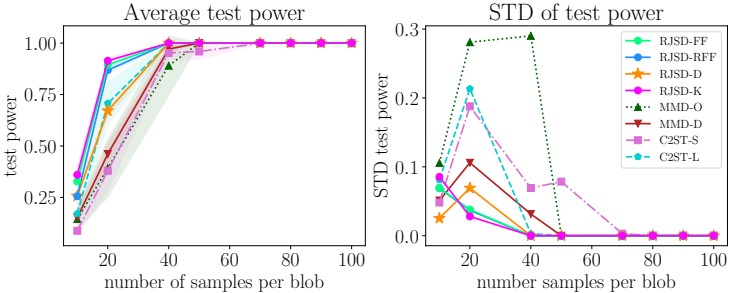

Figure 3: Power test for the blobs experiment.

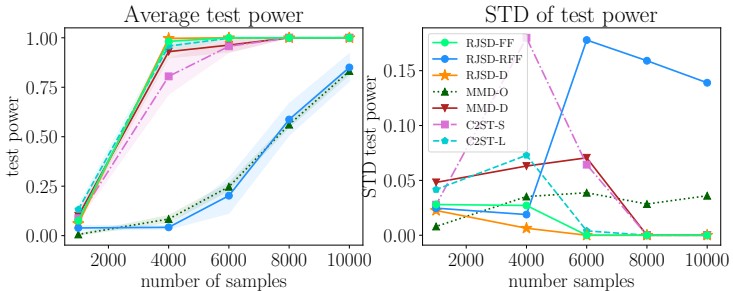

Figure 4: Power test for HDGM fixed N.

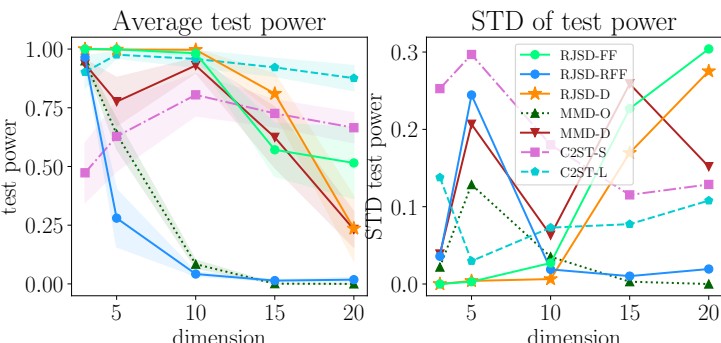

Figure 5: Power test for HDGM fixed d.

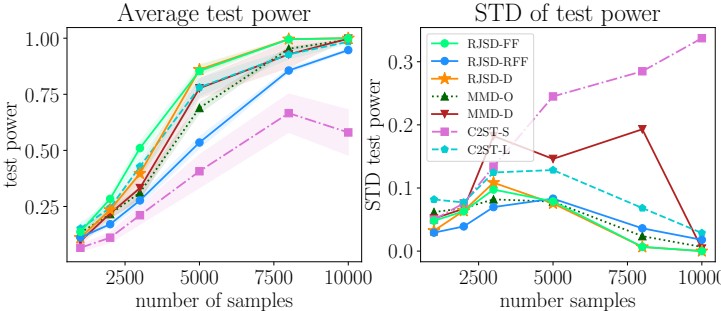

Figure 6: Power test for the Higgs dataset.