# OpenReview forum: "The Representation Jensen-Shannon Divergence"
_ICLR.cc/2024/Conference — Submitted to ICLR 2024_

### Official Review · Reviewer_apnH · 2023-10-24

**Soundness:** 3 good
**Presentation:** 3 good
**Contribution:** 2 fair
**Rating:** 5
**Confidence:** 3

**Summary:**

This paper studies kernel-based divergence and applications. Firstly, the probability distribution P is mapped to the non-central covariance matrix, C_P. Then, the corresponding probability density P_phi is defined from C_P. From such transformed probability distributions, the standard cross entropy and discrepancy measure called representation Jensen-Shannon divergence (RJSD) on probability distributions is defined. The authors investigated the relationship between discrepancy measures for C_P and that for P_phi. The random Fourier feature is employed to derive a tractable method of computing RJSD. The authors provide a theoretical analysis of the empirical estimation for the RJSD. Various numerical experiments indicate the statistical reliability of learning methods using the RJSD.

**Strengths:**

- The authors proposed a computationally efficient method to estimate the RJSD using the conjugate representation of the covariance matrix and the random Fourier feature.
- The authors established the theoretical analysis of the convergence rate of the proposed estimator for the kernel-based discrepancy measure.
- Some interesting and useful inequalities are derived.
- As applications of RJSD, the authors conducted various numerical experiments.

**Weaknesses:**

- The proposed divergence measure, RJSD, is regarded as a discrepancy measure for normalized kernel embeddings using the kernel function, kappa square. I think that there is no clear reason why the kernel embedding should be normalized to be a probability density. One could assess the discrepancy between unnormalized kernel embeddings by divergences for unnormalized functions, (pseudo-)distance measures, etc. What is the benefit of thinking of the normalization or the mapping to the set of probability densities for the non-negative kernel embeddings?

- In numerical experiments, the authors observed that learning algorithms based on the proposed divergence measure outperform existing methods. However, any insight into the numerical results is not sufficiently mentioned. For example, GAN using RJSD is found to be robust to the mode collapse. What makes the RJSD so reliable?

- The data size in numerical experiments is not necessarily large. When the proposed method is used for larger data set, how much is the computation cost?

**Questions:**

- The proposed divergence measure, RJSD, is regarded as a discrepancy measure for normalized kernel embeddings using the kernel function, kappa square. I think that there is no clear reason why the kernel embedding should be normalized to be a probability density. One could assess the discrepancy between unnormalized kernel embeddings by divergences for unnormalized functions, (pseudo-)distance measures, etc. What is the benefit of thinking of the normalization or the mapping to the set of probability densities for the non-negative kernel embeddings?

- In numerical experiments, the authors observed that learning algorithms based on the proposed divergence measure outperform existing methods. However, any insight into the numerical results is not sufficiently mentioned. For example, GAN using RJSD is found to be robust to the mode collapse. What makes the RJSD so reliable?

- The data size in numerical experiments is not so large. When the proposed method is used for larger data set, how much is the computation cost?

---

> ### Author Response · Authors · 2023-11-22
> **Rebuttal**
>
> Thank you for your valuable feedback and insightful comments on our paper. We appreciate the time and effort you have dedicated to reviewing our work. Next, we address the main concerns raised.
>
> **Questions**
>
> 1. We would like to clarify that the kernel used to measure RJSD does not need to be normalized to be a probability density, and that the only restriction is that $\kappa(x,x) = 1$. In definition 1. we normalize the kernel to be a probability density, to be able to compare the covariance operator entropy $S(C_\mathbb{P})$ to Shannon's entropy  $H(p)$. Notice that the result in Theorem 1. shows that the cross-entropy $H(p,\hat{p}) = S(C_\mathbb{P}) + \log(h)$, where $h$ is the normalization constant. However, $S(C_\mathbb{P})$ is independent of the normalization constant and does not require the kernel to be a density function to be computed. Additionally, notice that in Theorem 4 (Theorem 5 in the revised document), we establish the conditions under which $D_{JS}(\mathbb{P},\mathbb{Q}) = D_{JS}^\phi(C_\mathbb{P},C_\mathbb{Q}) $, and notice, that in this case, the normalization constant is not involved in this equivalence. In fact, one of the main benefits of the covariance operator entropy is that it avoids the intermediate step of density estimation.
>
> 2. The RJSD formulation has several advantages. First, in GANs, the problem of mode collapse is usually attributed to using a classifier to measure the divergence since the generator can learn to create a few realistic samples to fool the discriminator and trivially minimize the loss. Besides, since the discriminator handles each example independently, there is a lack of coordination between the gradients, and consequently, the generator is not penalized by producing similar samples. On the contrary, our framework processes the covariance matrices of entire batches of real and generated data rather than individual samples. This leads to coordinated gradients to either increase or decrease the divergence between the distributions. Second, RJSD avoids entropy regularizers to avoid mode collapse. Specifically, RJSD explicitly matches the entropy of the generated samples and the training distribution. This feature penalizes generative distributions with lower entropy than the real distribution.
>
> 3. In section 2.2 we mention that the cost of the kernel-based estimator is $\mathcal{O}(N^3)$, and the covariance-based is $\mathcal{O}(D^3)$. The cost of the kernel-based estimator is not really prohibitive, since we are able to compute eigendecompositions of up to 20-30K samples easily on modern computers. However, the covariance-based estimator is independent of sample size, which makes this estimator even more appealing, since even for larger amounts of data, we can estimate this quantity faster. In future work, we aim to test the proposed estimators in large scale datasets.
>
> We hope to have addressed your main concerns. If you have any further suggestions or questions, please feel free to let us know.

---

> > ### Comment · Reviewer_apnH · 2023-11-23
> >
> > Thank you for your responses and revisions.
> >
> > As for Question 1, I understand that one generally needs to compute the normalization constant to evaluate the RJDS except in exceptional cases, such as when the Gaussian kernel is employed. When we select kernel functions such as the Matern kernel according to prior knowledge of the smoothness of the probability density, I think the computation of the normalization constant, $h$, is required. If my understanding above is correct, my concern is still not resolved by the response from the author.
> >
> > In Question 2, the author provided some intuitive explanation of why the RJSD-based method can avoid mode collapse. In numerical experiments, the authors reported the superiority of the proposed method. On the other hand, which numerical results support the correctness of the intuition given by the author? Showing not only comparison results but also some ablation studies would be nice.

---

> > > ### Author Response · Authors · 2023-11-23
> > > **Further clarifications.**
> > >
> > > Q1. Thank you for your continued engagement with our work. We appreciate the opportunity to address your concerns regarding the computation of the normalization constant in the context of RJSD.
> > >
> > > RJSD is a measure of discrepancy between unit trace covariance operators in RKHSs, and it can be computed from samples using any positive definite kernel satisfying $\kappa(x,x) = 1$ for all $x \in \mathcal{X}$. The kernel-based estimator for RJSD is expressed as:
> > >
> > > \begin{equation}
> > >   D_{\scriptscriptstyle JS}^{\kappa}(\mathbf{X},\mathbf{Y}) = S\left(\mathbf{K_Z}\right) - \left(\tfrac{N}{N+M}S(\mathbf{K_X}) + \tfrac{M}{N+M}S(\mathbf{K_Y})\right),
> > > \end{equation}
> > > where $S(\mathbf{K_X}) = -\text{Tr}{\left(\tfrac{1}{N}\mathbf{K_X}\log \tfrac{1}{N}\mathbf{K_X}\right) }$. It's important to note that the computation of the normalization constant $h$ is not required for evaluating RJSD. The only requirement is that $\kappa(x,x) = 1$ for all $x \in \mathcal{X}$. Therefore, RJSD can be estimated using any valid kernel function (not only the Gaussian kernel) without requiring to compute  the normalization constant $h$.
> > >
> > > In our paper, we introduced the normalization constant $h$ uniquely aiming to explore the relationship between covariance operator quantities and classical measures like Shannon's entropy and Jensen-Shannon divergence. However, for the estimation of RJSD itself, the computation of this normalization constant is unnecessary and can be completely avoided.
> > > We hope this clarification addresses your concern.
> > >
> > > Q2. We appreciate your interest in the numerical support for the intuition presented in Question 2 regarding the avoidance of mode collapse.
> > >
> > > In our numerical experiments, we observed that the RJSD-GAN consistently demonstrated effectiveness in mitigating mode collapse. A key result supporting this observation is presented in Table 2, where RJSD outperformed several state-of-the-art GAN formulations on a challenging mode-collapse dataset (Stacked MNIST). Notably, RJSD achieved this without the need for additional entropy regularizers or specific mode-collapse prevention mechanisms beyond the learning function itself.
> > >
> > > Specifically, our approach, RJSD, successfully captured all 1000 modes in the dataset. This achievement is crucial in demonstrating that RJSD effectively matches the entropy of the generated samples with the training distribution. Furthermore, the steady generation of samples from all classes led to the lowest KL-divergence compared to baseline approaches. The ability of RJSD to capture all modes and minimize KL-divergence underscores its capability to match the distribution of generated samples with the training distribution. We acknowledge the importance of presenting not only comparison results but also conducting ablation studies, and we will pursue this goal in future work.

---

> > > > ### Comment · Reviewer_apnH · 2023-11-23
> > > >
> > > > Thank you for the reply. I'll scrutinize the paper again and reconsider the rating if necessary.
> > > > Best regards,

---

### Official Review · Reviewer_UrV9 · 2023-10-27

**Soundness:** 2 fair
**Presentation:** 2 fair
**Contribution:** 2 fair
**Rating:** 6
**Confidence:** 3

**Summary:**

This paper proposes a novel metric, the representation Jensen-Shannon divergence (RJSD), betweeen probability distributions. RJSD generalizes quantum Jensen-Shannon divergence to the covariance of a (potentially infinite-dimensional) RKHS embedding of the data. A few different estimators of RJSD are proposed. A simple experiment on synthetic data demonstrates the consistency of these estimators. Then, several experiments (on mostly synthetic data) demonstrate the utility of RJSD for (a) training GANs to avoid mode collapse and (b) nonparametric two-sample testing.

**Strengths:**

The paper presents a theoretically interesting new metric that relates RKHS embeddings with standard information-theoretic quantities. Experiments are performed on several different problems to illustrate the utility of this new quantity and its estimators.

**Weaknesses:**

**Major**

1. The paper frequently uses notation and terminology that has not been previously defined, and seems to assume some familiarity with quantum mechanics/quantum information theory. Reading this paper with backgrounds in information theory and in functional analysis, but not in quantum information theory, I think I was able to make the right educated guesses to understand most of what the paper is doing, but this guesswork shouldn't be necessary to read the paper, and I don't think this would be accessible to much of the ICLR community.
Some examples:
    1. Equation (2): What is $\otimes$? It seems like an outerproduct, but it's not clear.
    2. Bra-ket notation is used for inner products (e.g., in Definition 1 and Theorem 4) without any introduction. I happen to have heard of this notation, but have never seen it in a paper, and doubt many people in ICLR are familiar with it.
    3. Theorem 5: It's unclear what the norms $||\cdot||*$ and $||\cdot||_{\text{HS}}$ are.
    4. Section 3, just before Definition 2, "The Quantum counterpart of the Jensen-Shannon divergence (QJSD) between density matrices $\rho$ and $\sigma$ is defined as... where $S(\cdot)$ is von Neumann’s entropy.": What is a "density matrix"?
    5. Page 4, First Sentence: The univariate function $H$ is never defined; only the bivariate cross-entropy function was introduced in Eq. (6).

2. I don't understand the point of much of Section 2 (Background). For example, Theorems 1 and 2 are never referenced or related to the remainder of the paper. I suggest reducing some of this material to make space for (a) clear definitions of the notation used throughout the paper and (b) further motivation and details regarding the experiments in Section 5.

3. The abstract and contributions section both claim that the paper presents consistency and convergence rate guarantees for the proposed estimators, but I couldn't find any of these guarantees in the paper. The only guarantee presented (Theorem 2) applies only for finite-dimensional feature representations ($D < \infty$), so it doesn't seem to apply to the new estimators proposed.

4. Given the lack of supporting theoretical results (previous point), I was not very convinced by the experimental results. All of them are on simple, mostly synthetic datasets. While RJSD does perform well on many of the tasks, the improvements are mostly small, and, more importantly, there is no discussion of *why* RJSD might perform better than other metrics on these tasks.

5. Section 5.3 only presents the average power of various tests, and I didn't see any evidence that the tests obey their nominal significance levels.

**Minor**

1. Page 2, Second Bullet, "An estimator from empirical covariance matrices.... Additionally, an estimator based on kernel matrices... Consistency results and sample complexity bounds for the proposed estimator are derived.": It's unclear here which of the two estimators the consistency results and sample complexity bounds apply to. Please clarify the wording here.

2. Page 5, just after Theorem 5, "From this result we should expect RJSD to be at least as efficient as MMD for identifying discrepancies between distributions": To give contextualize this statement, perhaps it should be noted that, among proper metrics between probability distributions, MMD is quite weak [RRPSW15, SULLZP18].

**References**

[RRPSW15] Ramdas, A., Reddi, S. J., Póczos, B., Singh, A., & Wasserman, L. (2015, March). On the decreasing power of kernel and distance based nonparametric hypothesis tests in high dimensions. In Proceedings of the AAAI Conference on Artificial Intelligence (Vol. 29, No. 1).

[SULLZP18] Singh, S., Uppal, A., Li, B., Li, C. L., Zaheer, M., & Póczos, B. (2018). Nonparametric density estimation under adversarial losses. Advances in Neural Information Processing Systems, 31.

**Questions:**

**Minor**

1) Abstract: "Our approach embeds the data in an reproducing kernel Hilbert space (RKHS)... Therefore, we name this measure the representation Jensen-Shannon divergence (RJSD).": I didn't understand the use of "Therefore" here; i.e., how does the second sentence follow from the first?

2) Page 2, Last Sentence, "If we normalize the matrix $K_X$ such that, Tr$(K_X) = 1$...": There are many ways such normalization could be performed. Is this done by scaling (i.e., replacing $K_X$ with $\frac{K_X}{\text{Tr}(K_X)}$)? If so, this should be clearer.

3) Page 3, Last Two Sentences, "Let $\hat P_\gamma(x)$ be the empirical kernel density function by a Gaussian kernel with scale parameter $\gamma$. Dmitriev & Tarasenko (1974) demonstrate that $H(\hat P_\gamma)$ converges to $H(\mathbb{P})$ as both, $N$ and $\gamma \to \infty$, with probability one.": I was quite confused by the meaning of $\gamma$ here. Typically, for the kernel density estimate (KDE) to be consistent, the bandwidth (which I think of as synonymous with the "scale parameter") should shrink (approach $0$, not $\infty$). Does "scale parameter" here mean the reciprocal of the bandwidth (i.e., the "precision" of the Gaussian)? Also, are there no additional assumptions required here? For example, consistency of the KDE requires that the bandwidth $h$ not shrink too quickly (e.g., $N h^d \to \infty$).

4) Page 5, Equation (12): Typo: I think $D_{JS}$ should be $D_{JS}^\phi$, correct?

5) I don't understand why the exponential moving average (EMA) is applied in Section 5.1, as this doesn't tell us anything about the performance of RJSD and many of the baseline compared could similarly be modified to up-weight recent data.

6) How exactly is the RJSD metric utilized for two-sample testing in Section 5.3? Usually, either a permutation test or an analytically derived asymptotic null distribution is used, but I didn't see any explanation of this.

---

> ### Author Response · Authors · 2023-11-22
> **Rebuttal**
>
> Thank you for your valuable feedback and insightful comments on our paper. We appreciate the time and effort you have dedicated to reviewing our work. Next, we address the main concerns raised.
>
> **Major**
>
> 1. *Issues about confusing notation.* We appreciate your thorough review and feedback on the clarity of notation and terminology used in our paper. Your observations regarding the potential challenges for readers without a background in quantum mechanics/quantum information theory are valuable. In response to your concerns, we revised the manuscript to enhance the clarity of notation and provide more detailed explanations of relevant operations such as the tensor product $\otimes$, nuclear norm $\lVert \cdot \rVert_*$, Hilbert-Schmidt norm $\lVert \cdot \rVert_{HS}$, among others. Regarding the bra-ket notation, we changed it to a more familiar notation in Machine learning, using a more standard inner product instead avoiding any confusion among the readers. We also defined properly entropy and cross-entropy and added a footnote to define a density matrix.
>
> 2. Theorems 1 and 2 are important for establishing a connection between kernel-based entropy and Shannon's differential entropy. The convergence and consistency of RJSD are addressed later in the document where we leverage the results in Bach, (2022). We added the following:
> "Leveraging the convergence results in Bach, (2022)[Proposition 7] of the empirical estimator $S(\mathbf{K_X})$ to $S(C_{\mathbb{P}})$, we can show that $D_{\scriptscriptstyle JS}^{\kappa}(\mathbf{X},\mathbf{Y})$ converges to the population quantity $D_{\scriptscriptstyle JS}^\phi(C_{\mathbb{P}}, C_{\mathbb{Q}})$ at a rate $\mathcal{O}\left(\frac{1}{\sqrt{N}}\right)$, assuming $N=M$. Details of this rate are given in Appendix A.8. Additionally, a direct consequence of Theorem 2 is that, under the same assumptions of the theorem, $D_{\scriptscriptstyle JS}^{\kappa}(\mathbf{X},\mathbf{Y})$ converges to  $D_{\scriptscriptstyle JS}(\mathbb{P},\mathbb{Q})$ as $N\rightarrow \infty$ with probability one."
> We acknowledge that this might not have been expressed clearly, and we enhanced the clarity of our paper by explicitly stating these two different results.
> 3. This concern is related to the previous point. We acknowledge the potential for misunderstanding and appreciate the reviewer's feedback. This has been addressed in the revised manuscript.
> 4. Having shown the supporting theoretical results of the proposed RJSD estimator (Points 2 and 3),  we appreciate your feedback regarding the clarity of the performance improvement over other metrics. In response, we would like to highlight Theorem 5 (Theorem 6 in the revised manuscript) in our paper, where we establish that RJSD serves as an upper bound on MMD. This result implies that if MMD can effectively distinguish between two distributions, then RJSD is expected to have similar or superior discriminative power. We argue in the paper that this theoretical relationship suggests that RJSD should be at least as good as MMD in identifying discrepancies between distributions. While it is true that the observed performance gain may not bet substantial, deep kernel MMD is a state-of-the-art approach. In this context, even marginal improvements in performance on the evaluated datasets are noteworthy. To address your concern, in the revised paper, we elaborate further on why RJSD has a superior test power. We agree that these additional explanation will convey more clearly the advantages offered by our proposed method.
> 5. We appreciate your attention to the presentation of the test power results in Section 5.3 and your concern about the significance levels. To address this, we would like to clarify that all tests were conducted with a significance level of $\alpha = 0.05$. Detailed information about the testing procedure was provided Appendix C.3. Additionally, we included figures (3 to 6 in the appendix) displaying the standard deviations of test power for all experiments, offering a more comprehensive view of the test performance. Furthermore, we have made the code used to generate these results available on an anonymous GitHub repository to ensure transparency and reproducibility. This allows interested readers to replicate our experiments and verify the reported results. We believe these additional details and resources strengthen the robustness of our findings.

---

> > ### Author Response · Authors · 2023-11-22
> > **Questions**
> >
> > **Questions**
> > 1. The first sentence finishes saying that our metric uses the "covariance operator in this **representation** space". For this reason (**therefore**), we name our divergence, the **representation** Jensen-Shannon divergence.
> > 2. We agree that the normalization step should be clearer. As you pointed out the normalization is done by dividing over the trace ($\text{Tr}(\mathbf{K_X}) = N$ , because $\kappa(x,x) = 1 \forall x \in \mathcal{X}$), therefore, in the revised paper, we  make this normalization explicit in the kernel-based estimator as follows: $S\left(\mathbf{K}_{X}\right) = -\text{Tr}{\left(\frac{\mathbf{K}_X}{N}\log \frac{\mathbf{K}_X}{N}\right) } $.
> > 3. In our paper we use the definition of Gaussian kernel $\kappa(x,x') = \exp(-\gamma\lVert x - x' \rVert^2)$ and we regard $\gamma$ as the scale parameter. Here, we can see that $\gamma = \frac{1}{2\sigma^2}$, where $\sigma$ is the kernel bandwidth. Therefore, as $\gamma \rightarrow \infty $, $\sigma \rightarrow 0$, as you correctly mentioned. To avoid confusion, in the revised paper, we properly define the Gaussian kernel that we use and additional assumptions that guarantee the consistency of the KDE.
> > 4. It is correct, this is a typo. (Addressed in the revised manuscript)
> > 5. We use Exponential moving averages of the covariances to preserve information from past samples which smooths out the estimation as we can see in Fig. 1. This is something which cannnot be done with the kernel matrices, as we would require to compute pairwise similarities to all previous samples. In particular, Algorithm 2 in the appendix shows that given a new batch, we can obtain an updated covariance matrix that contains information from all samples in the past. This property allows the method to be scalable. For space reasons, we could not describe these benefits in more detail.
> > 6.  We would like to mention that in the main paper, page 8, last sentence, we refer readers to appendix C.3 where we explain the testing procedure. Specifically, we address the questions that you ask. Next, we quote appendix C.3.
> >
> > > The procedure is the following: for synthetic datasets, we create the sets $X_{train} \in \mathbb{R}^{N\times d}$ and $Y_{train} \in \mathbb{R}^{M \times d}$. Then, we learn the kernel/covariance/classifier for each of the methods on that training set. We then sample a testing set  $X_{test} \in  \mathbb{R}^{N\times d}$ and $Y_{test} \in  \mathbb{R}^{M\times d}$ and perform a permutation test. We compute the statistic on the testing set and perform 100 permutations to generate the surrogate of the distribution of the measurement under the null hypothesis. Finally, we compute the rejection threshold, and if the statistic is greater than this threshold we reject the null hypothesis. This is done for 100 independent testing sets. Finally, we repeat the experiment ten times and compute the average test power.
> >
> > We believe that the proposed enhancements will significantly improve the clarity and quality of the paper. If you have any further suggestions or questions, please feel free to let us know.

---

> > > ### Comment · Reviewer_UrV9 · 2023-11-22
> > >
> > > Thanks to the authors for their responses and revisions. The revisions, especially the new Theorem 3, address most of the issues I had with the paper. Accordingly, I am raising my score from 3 to 6.
> > >
> > > My main remaining concern is that it's still unclear why/when RJSD performs well in two-sample testing. Specifically, what kinds of alternatives is it most/least sensitive to? I don't find the inequality $D_{JS}^\phi \geq \frac{1}{8} MMD$ very compelling in this regard. What is important for two-sample testing is not only the actual value of the statistic, but also its null distribution. As an (admittedly very artificial) example, if $\lceil \cdot \rceil$ is the ceiling ("rounding up") function, then $\lceil MMD \rceil \geq MMD$, but $\lceil MMD \rceil$ is probably worse than MMD for two-sample testing, as it's just throwing away information.
> > >
> > > Some minor points remaining:
> > >
> > > 1.  Minor error in the last sentence of the proof of Theorem 3:
> > > > "Finally, since both series $\sum_{N = 1}^\infty \exp(-CN^{1/10})$ and $\sum_{D = 1}^\infty \exp(-CD)$ the theorem holds.
> > >
> > > This should probably say "since both series... **converge**". Also, I think this step is implicitly using the first Borel-Cantelli lemma; I suggest saying this explicitly.
> > >
> > > 2. I think it's still relatively unclear in the main paper why the exponential moving average (EMA) is applied in Section 5.1. Although the explanation that this can be implemented more efficiently for RSJD than for other approaches makes sense, if this explanation isn't clear to a reader of the paper, then I think it would be better to remove EMA from the paper, especially since RJSD already performs quite well without EMA.
> > >
> > > 3. To clarify my concern about significance levels: I understand that the tests were run with *nominal* significance level $\alpha = 0.05$, but the actual (empirical) significance level of a test may differ from its nominal level. Since the authors clarified that they used permutation tests, this is no longer much of a concern, as the actual significance level of a permutation test converges quickly to the nominal level as the number of permutations increases. One way to more clearly illustrate this would be to plot an ROC curve, where the $x$-axis is the actual (empirical) significance level and the $y$-axis is the actual (empirical) power. (I don't think this is important for this paper -- just trying to clarify what I was saying earlier.)

---

### Official Review · Reviewer_kvgr · 2023-10-30

**Soundness:** 2 fair
**Presentation:** 3 good
**Contribution:** 2 fair
**Rating:** 5
**Confidence:** 4

**Summary:**

In this paper, a new notion of diferegence between probability distributions is introduced.
This notion is similar to the well studied quantum version of Jensen-Shannon divergence.
The construction is as follows:
1)  A new entropy notion of a single distribution is defined by taking the von Neumann entropy of the
covariance matrix of the data in a given RKHS.
2) The Jensen-Shannon divergence of a pair of is based solely on the entropy notion, and so a corresponding notion can be defined based on the entropy above.

Two main things are proved:
1)  A relation between the new JS divergence and the classical (non quantum) one.
  In particular, the the classical dominates the new one, for any properly normalised kernels.
  Some cases of equality are outlined.
2) A sort of consistency result, for the case when the kernel is approximated (to reduce computation), based on existing results for the actual kernel and the existing approximation results.

Small dataset experiments are performed to demonstrate the usability of the new notion for data generation
and for two point tests.

**Strengths:**

This is mostly a well written and well executed paper.
The subject of developing diveregnce notions between probability distributions, which are useful and easily computable, is important throughout machine learning.
The particular notion introduced in this paper is natural and should be studied.

**Weaknesses:**

The paper would have been much stronger if the performance improvement over MMD had been shown more clearly.  As it stands, there is a rather small performance gain, and it is not completely clear why.

I also have a question on the theory.


Q1)

Regarding the theory, how important is the requirement in def. 2 that the kernel should satisfy $k(x,x)=1$?  On the one hand, it is easy to arrange. On the other hand, it seems that consistency results require **probability normalized** Gaussain kernels (i.e. integrate to 1, so the Parzen estimator would be a density), and these have $k(x,x) \righrarrow \infty$ with $\gamma \rightarrow \infty$.    Thus it appears the method would not compute the true  JSD.  Please comment.
 Note that MMD does not have such problems.

Q1.1)

I'm using $\gamma \rightarrow \infty$ above to be consistent with the paper. However, for standard definitions of scale, one should have $\gamma \rightarrow 0$ instead. Is this a typo? What definition of kernel and scale is being used?


Q2)


In the data generating experiment, the formulation (16) is not a GAN, as the authors note themselves.
To me, it is much more simular to a VAE, perhaps an Wasserstein VAE. Thus I'm not sure why a copmarison to GAN's is being made, these results do not seem to be  informative.  The comparson should be with VAE, and with MMD based ones such as InfoVAE.

Q3)


 Results on two sample tests:
In Table 3, there is no clear advatage of RJSD-D, over MMD-D, as the gaps at (400,500) are very small.

Results in Figure 4 do show an advantage of RJSD-D over MMD-D, but this is not a large advantage in almost all cases.  While it is interesing, this can hardly justify a new information notion. I'd suggest either performing additional experiments, or analysing the current ones in more detail, to see why RJSD performs better.


EDIT: The proof of Theorem 2 is vague and does not state clearly which result from  (Dmitriev & Tarasenko, 1974) is used. I believe their results can only be applied when the kernel is normalised. Since it is not normalised, Theorem 2 does not apply as stated.

**Questions:**

please see above

---

> ### Author Response · Authors · 2023-11-22
> **Rebuttal**
>
> Thank you for your valuable feedback and insightful comments on our paper. We appreciate the time and effort you have dedicated to reviewing our work.
>
> * *"The paper would have been much stronger if the performance improvement over MMD had been shown more clearly. As it stands, there is a rather small performance gain, and it is not completely clear why."* (**Q3**) We appreciate your feedback regarding the clarity of the performance improvement over MMD in our paper. We agree that providing a more explicit demonstration of this improvement is crucial and it would make the article stronger. In response, we would like to highlight Theorem 5 in our paper, where we establish that RJSD serves as an upper bound on MMD. This result implies that if MMD can effectively distinguish between two distributions, then RJSD can achieve at least similar discriminative power. We argue in the paper that this theoretical relationship suggests that RJSD should be at least as good as MMD in identifying discrepancies between distributions. While it is true that the observed performance gain may not be substantial, deep kernel MMD is a state-of-the-art approach. In this context, even marginal improvements in performance on the evaluated datasets are noteworthy. To address your concern, in the revised document, we elaborate further on why RJSD is expected to have a superior test power. We agree that these additional explanation will convey more clearly the advantages offered by our proposed method.
>
> * (**Q1**) The condition $\kappa(x,x) = 1$ ensures that the covariance operator will have unit trace. Note that from $\kappa(x,x)=1$, we obtain that $\lVert \phi(x)\rVert = 1$. Therefore, $\text{Tr}(\phi(x)\otimes \phi(x)) = \lVert \phi(x) \rVert^2$ for all $x \in \mathcal{X}$. On the other hand, in the results connecting kernel-based entropy with Shannon's differential entropy, as $\gamma \rightarrow \infty$, $\kappa(x,x)$ is still one, since for the Gaussian kernel $\kappa(x,x) = \exp(-\gamma\lVert x - x \rVert^2) = \exp(-\gamma \cdot 0) = 1 $ regardless the value of $\gamma$. To avoid confusion, we explicitly define the Gaussian kernel as above in the revised manuscript.
>
> * (**Q2**) We would like to clarify that our generative formulation is indeed a generative adversarial network (GAN). The formulation (16) is an adversarial game (min max problem), between the generator $G_\theta$ trying to maximize RJSD and the discriminator $\phi_\omega \circ f_\omega$ aiming to minimize RJSD. We mention, however, that we differ from the standard GAN formulation because instead of using a classifier as the discriminator, we use a Fourier Features network $\phi_\omega \circ f_\omega$ as the discriminator. Notice that in vanilla GANs, the loss function *implicitly* minimizes the Jensen-Shannon divergence; however, we achieve a similar goal by *explicitly* minimizing RJSD, without using a classifier.
>
> We believe that the proposed enhancements will improve the clarity and quality of the paper. If you have any further suggestions or questions, please feel free to let us know.

---

### Official Review · Reviewer_MfNe · 2023-10-30

**Soundness:** 3 good
**Presentation:** 3 good
**Contribution:** 2 fair
**Rating:** 6
**Confidence:** 3

**Summary:**

The paper introduces a novel divergence measure, coined representation Jensen-Shannon Divergence (RJSD). The contributions of the paper, as outlined by the authors, are primarily theoretical and can be summarized as follows:

* The introduction of a new type of divergence that avoids the need for density estimation.
* A method of estimating empirical covariance matrices needed for divergence computation.
* Establishing a connection between RJSD and other measures of discrepancy and tools in information theory.

The authors conducted a series of experiments to empirically demonstrate the usefulness of RJSD in constructing two-sample tests and in other machine learning applications.

**Strengths:**

The underlying mathematical concept of this divergence is intriguing and, as far as I know, original. In my opinion, the key insight can be easily grasped by contrasting different summary statistics derived from finite-dimensional kernels or random Fourier features. One common approach for comparing distributions involves evaluating empirical mean embeddings using L2 distance or calculating Hotelling's t-squared statistics. In contrast, the proposed method suggests using the traces of the covariance matrix of random features. While a significant portion of the paper is dedicated to demonstrating that this approach indeed results in a valid divergence, it intuitively makes sense that the trace can serve as a useful summary statistic.

**Weaknesses:**

A hard question seems to be why RJSD outperforms the "Learning Deep Kernels for Non-Parametric Two-Sample Test" experimentally or establishing the relationship between its power and power of MMD-like algorithms. To explore the latter, a promising starting point could be a comparison with "Interpretable Distribution Features with Maximum Testing Power," particularly without optimizing random Fourier features, employing identical random features for both tests and using only a small number of features.

Generally, I would appreciate a qualitative discussion around the situations in which a test relying on RSJD should be used over the alternative tests. Without practical guidance, this approach might be perceived as an academic novelty.

**Questions:**

See above

---

> ### Author Response · Authors · 2023-11-22
> **Rebuttal**
>
> Thank you for your valuable feedback and insightful comments on our paper. We appreciate the time and effort you have dedicated to reviewing our work.
>
> * *"A hard question seems to be why RJSD outperforms the "Learning Deep Kernels for Non-Parametric Two-Sample Test" experimentally or establishing the relationship between its power and power of MMD-like algorithms."* One important result in our paper is given in Theorem 5 (Theorem 6 in the revised manuscript). There, we show that RJSD is an upper bound on MMD, particularly, $D_{JS}^\phi(C_{\mathbb{P}},C_{\mathbb{Q}})\geq \frac{1}{8}\text{MMD}_{\kappa^2}(\mathbb{P},\mathbb{Q})$.  This result implies that if MMD can effectively distinguish between two distributions, then RJSD can achieve at least similar discriminative power. We argue in the paper that this theoretical relationship suggests that RJSD should be at least as good as MMD in identifying discrepancies between distributions. In the revised manuscript, we elaborate further on this theoretical property.
>
> We appreciate your suggestions of comparing our method against "Interpretable Distribution Features with Maximum Testing Power," and we will intend to perform additional experiments in the future work.

---

### Official Review · Reviewer_zuw1 · 2023-11-07

**Soundness:** 2 fair
**Presentation:** 1 poor
**Contribution:** 2 fair
**Rating:** 3
**Confidence:** 4

**Summary:**

This work introduces a divergence between probability distributions by embedding them into a Reproducing Kernel Hilbert Space (RKHS) called the Representation Jensen-Shannon Divergence (RJSD). The proposed divergence satisfies symmetry, positivity and boundedness and is well defined. The RJSD is upper bounded by the standard JSD, a well studied object in information theory. The RJSD corresponding to a chosen kernel K involves the von Neumann entropy of the integral operators corresponding to these kernels.

The authors propose a computable finite sample estimator for the RJSD which involves computing the von Neumann entropies of some kernel matrices (which are dual objects to the aforementioned integral operators). As such the estimator is sound, since concentration properties of kernel integral operators have been well studied [1]. Furthermore O(n^3) is not insurmountable for a first probe to consider the usefulness of the divergence object considered.

However the authors do not present any statistical properties of the estimator. Instead, they proceed to study a further approximation of this object based on Random Fourier Features (RFF) based approximates for the kernels. This approximation is somewhat limited in its scope, and does not particularly behave well when the problem dimensions is large. This is a serious confounder to the various conclusions drawn from later experiments in the paper, and needs sufficient ablation.

[1] On Learning with Integral Operators, Lorenzo Rosasco, Mikhail Belkin, Ernesto De Vito, JMLR 2010

**Strengths:**

The concept of the RJSD is novel and has not received sufficient attention so far. It is worth studying in its own right. The authors make a genuine effort to explore the relation of this object to previously known information theoretic quantities. The RJSD is also well estimable using concentration properties of the kernel integral operators and going forward may present a useful divergence between distributions.

The authors have established the relationships between the information theoretic objects well.

The experimental work in this paper is extensive.

**Weaknesses:**

The RFF based estimator is perhaps discussed too soon. I would have liked to read the properties and performance of the kernel matrix based estimator when O(n^3) is not really prohibitive. Typically we are able to compute eigendecompositions of upto 20-30K samples easily on modern laptops.

The authors say the estimator is "differentiable" in the introduction. Is this for the RFF estimator or the kernel matrix estimator. The differentiability for the kernel based estimator is not clear, and would be good to clarify.

The writing has a lot of room for improvement.
The following notational issues bothered (and confused) me while reading this paper.
1. curly brackets for {X, B_X} that defines measure space instead of the standard round brackets. Also appears above equation (8).
2. M_+^1(X) is complicated. Can you use a simpler notation say \mathcal{P} for the set of all distributions on (X, B_X)
3. is it true that phi(x) = K(x, .) since you are saying H is the RKHS corresponding to K. if phi is mapping into H then i think phi has to be K(x,.). If that is phi(x) is in some other Hilbert space H' (not necessarily the RKHS H) such that K(x,z) = <phi(x), phi(z)>_{H'} then is this sufficient. if by phi(x) you indeed meant K(x, .) then please clarify. This would then mean that C_P is the integral operator of the kernel which is a well studied object. (see [1] in summary), and provide some background on this operator.
4. "For the bounded kernel ... and trace class", is the boundedness only needed for the trace class? If so split this sentence into 2.
5. Is E[ K(X, X)] < inf necessary for the relation E[f(X)] = <f, mu_P>. I think it would hold even otherwise, just that both sides would be infinite.
6. In section 2.2 you say unit trace. Is this an assumption for everything that follows? Should all kernel matrices also be normalized to have unit trace? Regardless, if it is clear (or even nontrivial) as to what happens to the entropy when the trace is not unity, a clarification would be useful to the reader. I think this is the general relation: S(c * C_P) = c * S(C_P) + c * log (c). Please recall the fact that trace=1 is assumed anywhere else in the paper where that is used. Is the RJSD invariance to the scale of the kernel, eg if I use K1 = c * K, the RKHSs for K1 and K are the same but their norms are scaled versions of each other. How does this change the RJSD, if it doesnt make this clear.
7. In paragraph on covariance based estimator. Is this phi same as the phi? If not please use a different letter. Next please write an equation that describes a shift invariant kernel. Unless that is used the formula for p(w) using Bochners thoerem is not clear because till then kappa is a bivariate function. Use a different function name in that formula to avoid confusion. Also state that w_i are iid samples from p(w). Also use a different letter for phi_w since phi is already used in Section 2.1 for something else. State at the end of this paragraph how to choose D (perhaps point to Theorem 2)
8. Formally define entropy H and cross-entropy H(P,Q) otherwise Theorem 1 doesnt make any sense. Also is this differential entropy, please state this clearly. Differential entropy doesnt enjoy all the properties that entropy from information theory does. eg it can be negative.
9. In definition 1 you use K(x,x)=1 but that is not true when you are using trace(C_P) = 1. Please make sure things are consistent here.
Next in this definition, please use a different letter than P for the LHS. It is very confusing to see P both on the left and right side. This issue complicates things further when you are assuming some phi* exists in Theorem 3. Theorem 3 was incomprehensible and needs a significant edit after definition 1 has new notation for the LHS.
10. In Theorem 3 which space does phi* lie in. Please again use a different letter here. phi has been overloaded by being used 3 times at this point in the paper.
11. Theorem 5 has a ||.||* which is undefined. Here again you are using K(x,x)=1, does that clash with trace=1. Also formally define MMD_K clearly before stating the lemma. I feel Theorem 5 should just have 2 objects RJSD >= MMD. the other objects can be pushed to the proof.
12. Avoid using pi1 and pi2, it would perhaps be easier to read with the N/(N+M).
13. Section 4, is the omega in phi_w o f_w same for phi and w? It seems weird that the networks earlier and last layers have the same parameterization
14. Figure 1, please mention that the black line is for the closed form expressions between the Cauchy distributions. Please also mention in the caption why the steps are appearing, ie, because you are manually changing the parameters of the distributions at those points. Related to this, please give full forms of the different estimators you have used.
15. Figure 3, doesnt really add much value as such. What are we supposed to make of these letters? Are they good?

The comment about the kernel learning strategy is important, but mentioned in a fleeting manner. Please either elaborate on this or avoid it.

**Questions:**

Added to weaknesses box

---

> ### Author Response · Authors · 2023-11-22
> **Rebuttal**
>
> Thank you for your valuable feedback and insightful comments on our paper. We appreciate the time and effort you have dedicated to reviewing our work. Next, we address the main concerns expressed in the revision.
>
>
> * *"However the authors do not present any statistical properties of the estimator"*.
> We would like to clarify that we do introduce statistical properties of our estimator. The estimator of RJSD is based on the kernel-based estimator of Shannon’s entropy whose statistical properties are presented in Theorems 1 and 2 in the original paper. Specifically, we show in Theorem 1, that our entropy estimator can be related to a plug-in Parzen density estimator, with convergence and consistency extensively studied (Dimitriev \& Tarasenko ,(1974)) as we point out in the document.
> Moreover, we show in Theorem 2 a similar convergence result when using Fourier Features. Therefore, the statistical properties of RJSD are a direct consequence of these results. Notice, that these results apply, for both estimators, using Kernel matrices or the explicit covariance matrices. We acknowledge the potential for misunderstanding and appreciate the reviewer's feedback. To address this concern, in the updated draft we incorporated an additional theorem (New Theorem 2) explicitly outlining the convergence of the kernel-based and covariance-based entropy estimators to Shannon's differential entropy. Additionally, we added a paragraph after Eqn. 13 to discuss the convergence of the kernel-based RJSD estimator to the population quantity.
>
> * *"The RFF based estimator is perhaps discussed too soon. I would have liked to read the properties and performance of the kernel matrix based estimator when $O(N^3)$ is not really prohibitive."*
> We appreciate your suggestion to elaborate further the properties of the kernel matrix-based estimator. It's important to note that the introduction of the Fourier features approach is not solely aimed at reducing computational cost by approximating the kernel-based estimator. Instead, the Fourier features serve a dual purpose in our methodology. Firstly, they facilitate the construction of empirical covariance matrices, crucial for estimating RJSD. Secondly, these features enable a parameterization of the representation space, leading to a kernel-learning strategy. We treat the Fourier features as learnable parameters within a neural network (Fourier Feature network), optimizing them to maximize divergence and enhance the discriminatory power between sets of samples. Consequently, the Fourier features approach offers a more versatile estimator that extends beyond addressing computational cost.
> In the revised manuscript, we provide a clearer exposition of these two functionalities to enhance understanding. Also, we changed the order of the RJSD estimators, presenting first the kernel-based estimator as you suggested.
> * *"The authors say the estimator is "differentiable" in the introduction. Is this for the RFF estimator or the kernel matrix estimator. The differentiability for the kernel based estimator is not clear, and would be good to clarify."*
> We would like to clarify that when we mention the estimator as 'differentiable,' we are referring to both kernel-based estimator and the covariance-based estimator using Fourier Features. The kernel-based estimator is indeed differentiable, and this was introduced in Sanchez Giraldo & Principe (2013); Yu et al. (2021). On the other hand, Sriperumbudur & Szabó (2015) show that the Fourier Features are twice differentiable and therefore the kernel or matrix produced by this approach. Therefore, the result in Sanchez Giraldo & Principe (2013); Yu et al. (2021) holds for this estimator as well. We acknowledge that this may have been unclear in our original presentation. In the revised manuscript, we provided additional clarification on the differentiability of the kernel-based
> estimator, ensuring a more explicit and transparent discussion.
> * *"The writing has a lot of room for improvement. The following notational issues bothered (and confused) me while reading this paper."* Regarding the notational issues, we appreciate the thorough revision of the notation inconsistencies or confusing symbols, and we addressed all the concerns raised in the revised manuscript. Next, we reply individually to some of them that require further clarification.

---

> > ### Author Response · Authors · 2023-11-22
> > **Notational issues**
> >
> > 1. Addressed.
> > 2. Addressed.
> > 3. Yes, $\phi(x) = \kappa(x,\cdot)$, and we expressed this relationship through the inner product as $\kappa(x,x')= \langle\phi(x),\phi(x')\rangle_\mathcal{H}$.
> > 4. Addressed.
> > 5. $E[ \kappa(X, X)] < \infty$ is a necessary condition as it is shown by Smola et al. (2007) and Gretton
> > et al. (2012)[Lemma 3]. Particularly, $E[\kappa(X, X)] < \infty$ is the sufficient condition to ensure that $\mu_{\mathbb{P}}$ is an element of the Hilbert space $\mathcal{H}$ which allows the inner product $\langle f, \mu_P \rangle$.
> > 6. The unit trace of the covariance operator is a necessary assumption so that its eigen-values sum-up to one and can be used to compute the entropy. Assuming $\kappa(x,x) = 1 \forall x$ guarantee that the trace of the covariance operator is one  (Bach, 2022). Therefore, as you pointed out, all kernel matrices should be normalized to have unit trace. That's why we state that "If we normalize the matrix $\mathbf{K_X}$ such that, $Tr(\mathbf{K_X}) = 1$, $\mathbf{C_X}$ and $\mathbf{K_X}$ have the same non-zero eigenvalues (Sanchez Giraldo et al., 2014)". For the sake of clarity, we made this normalization explicit as follows: "We focus on the case of normalized kernels where $\kappa(x, x) = 1$ for all $x\in \mathcal{X}$. We denote the Gram matrix $\mathbf{K_X}$, consisting of all normalized pairwise kernel evaluations of data points in the sample $\mathbf{X}$, that is $(\mathbf{K_X})_{ij} = \kappa(x_i, x_j)$ for $i, j = 1, \dots, N$. It can be shown that $\mathbf{C_X}$ and $\frac{1}{N}\mathbf{K_X}$ have the same non-zero  eigenvalues (Sanchez Giraldo et al., 2014; Bach, 2022), yielding the kernel-based entropy estimator:
> > $ S\left(\mathbf{K_X}\right) = -\text{Tr}{\left(\tfrac{1}{N}\mathbf{K_X}\log \tfrac{1}{N}\mathbf{K_X}\right) }
> > $."
> > 7. $\phi$ is a general notation to describe any mapping (function) from $\mathcal{X}$ to $\mathcal{H}$. Therefore, we use $\phi_\omega$ to denote a mapping that is parameterized by $\omega$ (Random Fourier Features mapping).
> > 8. Addressed.
> > 9. * Remember that we mention that the kernel matrix should be normalized so that $\text{Tr}(\mathbf{K_X}) = 1$. Since $\kappa(x,x) = 1$, the kernel matrix should be normalized by the number of samples to ensure unit trace. For the sake of clarity, we express the kernel estimator in terms of $\mathbf{K_X}/N$.
> >     * With respect to definition 1, we use $\mathbb{P}_{\phi}$ to denote the approximation to $\mathbb{P}$ given by a kernel density estimator with a mapping $\phi(x) = \kappa(x, \cdot)$. To make this less confusing, we denote the density function as $p(x)$ and its approximation as $\hat{p}(x)$.
> > 10. To ensure consistent notation we use the standard notation $\phi(\cdot)$ to refer to any mapping to some RKHS $\mathcal{H}$. The reason we use $\phi^*(\cdot)$ is to denote some specific mapping $\phi$ that estimates the density function $p(x)$. If this is the case, RJSD is equal to the true Jensen Shannon divergence.
> > 11. Addressed
> > 12. Addressed
> > 13. In section 4, $\omega \in \Omega$ refers to all parameters in the network. Notice, that the Fourier features are treated as parameters of a neural network that can be trained. Essentially, the Fourier features can be seen as a linear layer with cosine and sine activation functions. Here, all parameters are optimized jointly through backpropagation.
> > 14. Addressed
> > 15. Figure 3 could be removed. We just wanted to show evidence that the RJSD-GAN was producing actual digits and that they were visually similar to the ones in the stacked MNIST dataset.
> >
> >
> > We believe that the proposed enhancements will significantly improve the clarity and quality of the paper. If you have any further suggestions or questions, please feel free to let us know.

---

### Author Response · Authors · 2023-11-22
**Response to reviewers and rebuttal revision**

We thank all the reviewers for your dedicated time and effort in reviewing our paper. We are encouraged that the reviewers find our article well written and well executed, as well as considering the representation Jensen-Shannon divergence (RJSD) to be an academic novelty, theoretically interesting, useful, easily computable, and well estimable. We appreciate your thoughtful comments, suggestions, and positive feedback. Accordingly, we have updated the draft of the article to incorporate the suggestions of the reviewers. Specifically, we have adjusted the notation to make the work more accessible. Also, we emphasize the statistical properties (convergence results) of the RJSD estimators, both kernel-based and covariance-based. Additionally, we elaborate on why RJSD is expected to have a higher discriminatory power than MMD. We believe that the proposed suggestions have significantly improved the clarity and quality of the paper.

Next, we individually answer specific questions to all reviewers.

---

### Meta-Review · Area_Chair_Pn1s · 2023-12-10

**Metareview:**

In this paper, the authors introduce a kernel algorithm to estimate the Shannon-Jensen divergence. The reviewers all see merits in the proposed method. However, they all indicate the paper is not ready for publication. First, the paper is hard to follow and introduces different concepts without clearly defining them. The authors need to improve how they communicate their results.

The authors present Theorem 6 as proof that RJSD is superior to MMD, but showing the RJSD is larger than MMD for two distributions does not prove it is a better estimate, as the variance for the estimator for the same distribution plays a crucial role.

Finally, the experimental results are not very compelling. The results for GANs are old, and the field has moved considerably since the inception of standard GANs or DCGANs. Also, the datasets are not very representative. The comparison with MMD for the two sample tests is not very positive. The problem with the experiments highlights the second problem with the paper: there is no rationale for this new divergence. Besides proposing a novel estimator, it would be interesting to have a justification for when this estimator is better than the ones we already have.

**Justification For Why Not Higher Score:**

The paper could be accepted, even when the scores are low. The authors have interesting contributions. In any case, I think the paper can be improved significantly and be submitted next year (or to ICML).

**Justification For Why Not Lower Score:**

not applicable

---

### Decision · Program_Chairs · 2024-01-16

Reject